# Human non-canonical inflammasomes activate CASP3 to limit intracellular *Salmonella* replication in macrophages

**Madhura Kulkarni‡, Christopher M. Bourne‡, Ashutosh B. Mahale, Patrick M. Exconde, Cecelia Murphy, Haley T. Goodrow, Sofia Cervantes, Matilda Kardhashi, Mirai Kambayashi, William Yoo, Tristan J. Wrong, Robert C. Patio, Bohdana M. Discher, Cornelius Y. Taabazuing** *

Department of Biochemistry and Biophysics, University of Pennsylvania Perelman School of Medicine, Philadelphia, Pennsylvania, United States of America

‡ MK and CMB are co-first authors.
* Cornelius.Taabazuing@pennmedicine.upenn.edu

## Abstract

Inflammasomes are signaling platforms that activate inflammatory caspases to initiate innate immune responses. Canonical inflammasomes sense diverse threats and activate CASP1, which cleaves the pro-inflammatory cytokines IL-1β and IL-18 and the pore-forming protein gasdermin D (GSDMD) to induce pyroptosis. In contrast, the non-canonical inflammasome senses bacterial lipopolysaccharide (LPS) through CASP4 and CASP5 to induce pyroptosis. While CASP1 substrates are well defined, those of CASP4 and CASP5 remain less understood. Here, we show that intracellular LPS and the gram-negative bacterial pathogen *Salmonella* activate CASP4/5 in macrophages to directly cleave and activate CASP3 and CASP7. Activated CASP3 subsequently cleaves gasdermin E (GSDME). Surprisingly, CASP3, but not GSDME, was required for restricting intracellular *Salmonella* replication, suggesting a protective role for apoptotic signaling. We further find that most GSDMD cleavage during non-canonical signaling is mediated by CASP1. Consistent with this, loss of GSDMD, but not GSDME, reduced LDH release, establishing GSDMD as the primary driver of pyroptosis during LPS transfection. In contrast, during *Salmonella* infection, cell lysis occurred independently of both GSDMD and GSDME, suggesting the involvement of alternative lytic mechanisms. Finally, we demonstrate that CASP4/5 activation of CASP3/7 and GSDME occurs in human primary macrophages, defining CASP4/5 as dual apoptotic initiator and inflammatory caspases in innate immunity.

## Author summary

The immune system uses specialized sensors called inflammasomes to detect infection and trigger protective responses. These responses are carried out by

**Data availability statement:** All data are available in the manuscript or in the Supporting information.

**Funding:** This work was supported by the National Institute of General Medical Sciences 1R35GM155239-01 to CYT, by The Colton Center for Autoimmunity at Penn to CYT, by the Burroughs Wellcome Fund 1054907 to CMB, and by the Penn Provost Postdoctoral Fellowship to CMB. The funders had no role in study design, data collection and analysis, decision to publish, or preparation of the manuscript.

**Competing interests:** The authors have declared that no competing interests exist.

enzymes known as caspases, which can induce an inflammatory form of cell death called pyroptosis to alert the immune system of a threat. The canonical inflammasome pathway activates caspase-1 (CASP1), whereas the non-canonical pathway activates caspase-4 and caspase-5 (CASP4/5). In this study, we found that non-canonical inflammasomes activate a different set of caspases, caspase-3 and caspase-7 (CASP3/7), which are typically associated with apoptosis, a form of cell death often considered non-inflammatory. We found that multiple pathways can drive cell death during infection and that activated CASP3 plays an important role in limiting the growth of *Salmonella* inside human immune cells. The activation of gasdermin E, a substrate of CASP3 that can convert apoptosis into pyroptosis, was not required for this protective effect. Together, our results reveal that inflammasome-driven immune responses are more flexible than previously appreciated and can engage multiple strategies to control infection. This work provides new insight into how immune cells balance inflammation and cell death to defend against bacterial pathogens.

## Introduction

Detection and elimination of invading pathogens by the immune system is essential for host survival. The immune response to invading pathogens must be tightly regulated to ensure effective pathogen clearance without excessive inflammation, which can lead to sepsis. Sepsis is a life-threatening condition that primarily affects children under the age of five and remains a leading cause of death worldwide [1,2].

A key component of the host response to invading pathogens is the assembly of multi-protein signaling complexes known as inflammasomes [3]. Inflammasome assembly activates cysteine proteases called caspases that induce inflammation and lytic cell death known as pyroptosis, a process critical for infection control [4,5]. The canonical inflammasome pathway activates caspase-1 (CASP1), whereas the non-canonical pathway activates caspase-4 and caspase-5 (CASP4/5) in humans and caspase-11 (CASP11) in mice, collectively referred to as inflammatory caspases [6,7]. CASP1 can be activated by diverse stimuli, including host or pathogen proteases and potassium efflux [8–11]. In contrast, CASP4/5 and CASP11 are activated by intracellular lipopolysaccharide (LPS) from Gram-negative bacteria such as *Salmonella enterica* serovar Typhimurium (*Stm*), an intracellular pathogen and a major cause of food-borne disease worldwide [6,12–14]. While CASP1 is important during early infection, CASP4 plays a critical role in restricting intracellular *Stm* replication at later stages of infection in human macrophages [15]. However, the mechanism underlying this late-stage restriction remains unclear.

The substrates of inflammatory caspases are key mediators of host defense, but their full range and functional significance remain incompletely defined [16]. In human cells, inflammatory caspases cleave the interleukin family of cytokines, IL-1β and IL-18, and the pore-forming protein gasdermin D (GSDMD) to induce pyroptosis [17–23]. Through this mechanism, pyroptosis promotes pathogen clearance and protects the host from infection [24–28]. In the absence of GSDMD, CASP1 can

instead process the apoptotic executioner caspases CASP3 and CASP7 to induce apoptosis [29]. A proteomic screen also identified CASP7 as a substrate of inflammasome-activated CASP1 in mice, although CASP3 activation in that study appeared to be independent of CASP1 [30]. Subsequent studies demonstrated that CASP1 can promote CASP3 activation in the absence of GSDMD during infection with *Stm* or stimulation with poly(dA:dT) [31,32]. This process depends on CASP1-mediated cleavage of Bid (BH3-interacting domain death agonist), which triggers CASP9-dependent activation of CASP3 [31,32]. These findings indicate that CASP1 can initiate apoptosis through both direct and indirect mechanisms.

Apoptosis is a regulated form of cell death that maintains tissue homeostasis and is generally considered immunologically silent. However, emerging evidence suggests that apoptosis contributes to host defense during infection [33–36]. For example, dendritic cells activate both pyroptosis and apoptosis during late stages of infection and this is essential for restricting *Legionella pneumophila* replication [35,36]. Similarly, activation of apoptotic signaling during viral infection can lead to CASP3-dependent cleavage of gasdermin E (GSDME), inducing pyroptosis and protecting human skin organoids from viral replication [37,38]. Whether apoptosis or CASP3 and GSDME-mediated pyroptosis broadly contributes to host defense remains unclear. Furthermore, although CASP1 is known to engage apoptotic pathways, it is not known whether human non-canonical inflammasomes (CASP4/5) similarly activate apoptotic caspases.

Substrates of the non-canonical inflammasome pathway are less well defined. A recent proteomic screen identified CASP7 as a substrate of CASP4 in humans, although its functional significance was not determined [39]. We and others recently reported that IL-1β and IL-18 are substrates of CASP4/5 [20–22]. Notably, *CASP4/5*-deficient THP-1 cells failed to activate CASP3 in response to intracellular LPS [21], suggesting that the human non-canonical inflammasomes may engage apoptotic signaling. However, whether this reflects direct cleavage of CASP3 and its functional relevance for host defense remain unknown.

In this study, we identify CASP3 and CASP7 as direct substrates of human inflammatory caspases. We show that CASP4/5 directly cleave CASP3/7, and that active CASP3 subsequently cleaves GSDME following stimulation by intracellular LPS and *Stm* infection. Although CASP3-dependent GSDME-mediated pyroptosis has been described in the context of chemotherapy and cytotoxic lymphocyte activity [40], it has not previously been linked to non-canonical inflammasome signaling. We further demonstrate that CASP3/7 activation and GSDME processing occur in primary human macrophages in response to intracellular LPS and *Stm*. Remarkably, CASP3, but not GSDME, was required to restrict intracellular *Salmonella* replication. These findings indicate that non-canonical inflammasomes contribute to host defense, in part by activating the apoptotic executioner caspases.

## Results

### Human inflammatory caspases directly cleave and activate executioner CASP3 and CASP7

Because previous work has shown that CASP1 can cleave CASP3/7, and CASP4 has been implicated in CASP7 processing [39], we first investigated whether inflammatory caspases directly target CASP3 and CASP7. Caspases cleave peptide bonds at aspartic acid (D) residues [41,42]. During apoptosis, initiator caspases CASP8 and CASP9 activate executioner caspases by cleaving CASP3 at D175 and CASP7 at D198, generating catalytically active large and small subunits [30,43]. We therefore hypothesized that inflammatory caspases also cleave CASP3 and CASP7 at these canonical sites. To test this, we first expressed human GSDMD as a control in HEK 293T cells and incubated lysates with recombinant active CASP1, CASP4, CASP5, CASP8, or mouse CASP11 for 1 or 3 hours. As expected, the inflammatory caspases efficiently processed GSDMD at both time points, confirming their catalytic activity (Fig 1A). We also expressed and purified C-terminally flag-tagged catalytically inactive CASP3 (CASP3$^{C163A}$) or CASP3$^{C163A}$ with a D175A mutation, as well as catalytically inactive CASP7 (CASP7$^{C186A}$) or CASP7$^{C186A}$ with a D198A mutation from HEK 293T cells. These proteins were incubated with recombinant active CASP1, CASP4, CASP5, or mouse CASP11 for 1 hour. CASP1, CASP4, CASP5 and to a lesser extent, CASP11, processed CASP3$^{C163A}$ and CASP7$^{C186A}$ but failed to process the D175A and D198A mutants

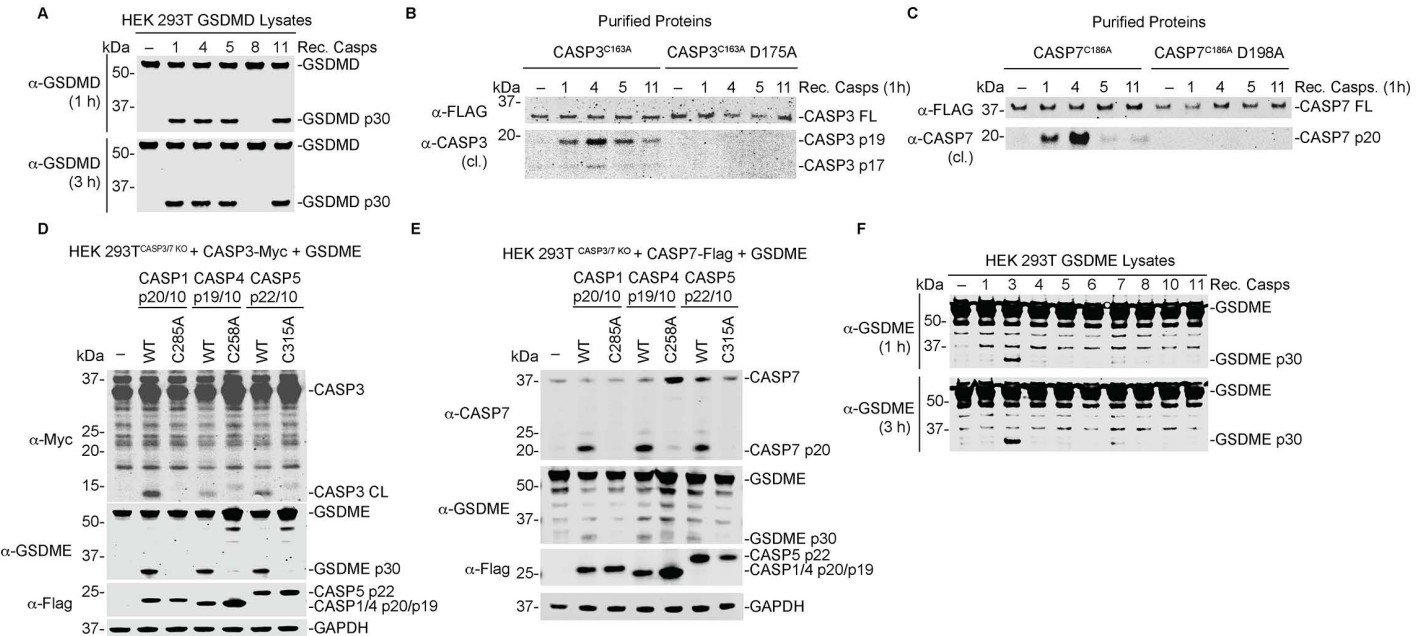

**Fig 1. Human inflammatory caspases directly cleave and activate executioner CASP3 and CASP7. (A-C)** HEK 293T cells were transiently transfected with plasmids coding for human GSDMD **(A)**, CASP3 **(B)**, or CASP7 **(C)**. Lysates **(A)** or purified CASP3 **(B)** or CASP7 **(C)** were then incubated with 0.25 activity units/µL of each indicated recombinant caspase for the specified times then cleavage products were analyzed by immunoblotting. **(D,E)** *CASP3/7* double KO HEK 293T cells were co-transfected with constructs encoding active CASP3 **(D)** or CASP7 **(E)**, and GSDME, and either catalytically active or inactive CASP1, CASP4, or CASP5 for 24 h then lysates were analyzed by immunoblotting. **(F)** HEK 293T cells were transiently transfected with a plasmid coding for GSDME for 24 h. Lysates were incubated with 0.25 activity units/µL of each recombinant caspase for 1 or 3 h, followed by immunoblot analysis. Data are representative of three or more independent experiments.

(Fig 1B and 1C). CASP5 also weakly processed CASP7 in this assay (Fig 1C). These results indicate that D175 and D198 are the cleavage sites for CASP3 and CASP7 respectively.

We next examined whether inflammatory caspases cleave CASP3/7 in cells. We generated *CASP3/7* double knockout (KO) HEK 293T cells (S1A Fig) and co-transfected these cells with either active or catalytically inactive inflammatory caspases together with catalytically inactive CASP3^C163A or CASP7^C186A. Consistent with our *in vitro* results, all inflammatory caspases processed CASP3^C163A and CASP7^C186A (S1B and S1C Fig). This processing depended on the activity of the inflammatory caspases and not on the activity of CASP3 or CASP7.

We further validated these findings using an orthogonal activation system. In this system, the caspase activation and recruitment domain (CARD) is replaced with a dimerizable (DmrB) domain, allowing selective activation of CASP1, CASP4, or CASP5 using the small molecule AP20187 [21]. HEK 293T cells stably expressing a DmrB-caspase and IL-18, a known substrate of inflammatory caspases, were transfected with CASP3 or CASP7. Treatment with AP20187 induced activation of DmrB-CASP1, CASP4, and CASP5, as indicated by IL-18 cleavage (S1D-S1I Fig). Consistent with our biochemical assays, activation of these caspases led to processing of both CASP3 (S1D-S1F Fig) and CASP7 (S1G-S1I Fig). These results demonstrate that inflammatory caspases cleave CASP3 and CASP7 upon activation in cells.

Processing of CASP3/7 in cells generated cleavage fragments that were detected by antibodies raised against activated species of CASP3 or CASP7, suggesting that inflammatory caspase-mediated cleavage of CASP3/7 results in activation. To test this directly, we co-expressed catalytically competent CASP3 (Fig 1D) or CASP7 (Fig 1E) with GSDME, a known CASP3 substrate [44], together with active or inactive inflammatory caspases in *CASP3/7* KO HEK 293T cells.

Active CASP3, and to a lesser extent CASP7, induced GSDME cleavage, indicating that inflammatory caspases cleave and activate CASP3/7.

To determine whether inflammatory caspases could directly cleave GSDME, we performed recombinant caspase cleavage assays. Lysates from GSDME-expressing HEK293T cells were incubated with apoptotic caspases (CASP3, CASP6, CASP7, CASP8) or inflammatory caspases (CASP1, CASP4, CASP5, CASP11) (Fig 1F). CASP3 robustly cleaved GSDME, whereas CASP7 produced only minimal cleavage after prolonged incubation, suggesting that CASP7-mediated GSDME processing is likely not physiologically significant. In contrast, inflammatory caspases did not directly cleave GSDME. Collectively, these findings demonstrate that human inflammatory caspases CASP1, CASP4, and CASP5 directly cleave CASP3 and CASP7 at their canonical activation sites and promote their activation. Among the caspases, CASP3 appears to be the primary mediator of GSDME cleavage.

## CASP1, CASP8, and CASP9 are not required for CASP3/7 and GSDME processing during non-canonical inflammasome activation by intracellular LPS

Our in vitro and cellular assays showed that inflammatory caspases cleave and activate CASP3/7, and that CASP3 can subsequently cleave GSDME to drive pyroptosis [40]. Although CASP3-mediated GSDME cleavage has been shown to induce pyroptosis, its contribution to non-canonical inflammasome mediated cell death remains unclear. To address this, we transfected LPS into PMA-differentiated THP-1 macrophages to activate CASP4/5 (Fig 2A). CASP1 can be activated downstream of CASP4/5 through NLRP3 inflammasome signaling triggered by potassium (K+) efflux via GSDMD pores [8,9,45]. Prior work has shown that both GSDMD and GSDME contribute to cell death and cytokine release during CASP1 activation [46]. To determine whether CASP1 contributes to CASP3/7 activation downstream of CASP4/5, we pretreated THP-1 cells with the NLRP3 inhibitor MCC950 [47] prior to LPS transfection (Fig 2A). LPS stimulation induced CASP3, CASP7, and GSDME cleavage, along with robust CASP1 activation and GSDMD cleavage, as indicated by the appearance of the CASP1 p10 subunit and GSDMD p30 fragment (Fig 2A). MCC950 pretreatment abolished CASP1 activation and reduced GSDMD processing. Notably, MCC950 pretreatment increased CASP3, CASP7 and GSDME cleavage compared to LPS alone, suggesting that CASP4/5 drive enhanced CASP3/7 and GSDME activation in the absence of CASP1 activation. Consistent with this, a cell permeable CASP3/7 activity detection reagent showed increased CASP3/7 activity in MCC950-pretreated cells relative to control and LPS-treated cells (Fig 2C). Despite the loss of CASP1 activation, the kinetics of Sytox Green uptake were unchanged, likely reflecting increased GSDME processing and pore formation (Fig 2B).

In addition to CASP1, CASP8 and CASP9 are known to activate CASP3/7 during apoptosis, and can contribute to CASP3-dependent cleavage of GSDME to induce pyroptosis [44,48]. We therefore examined the contribution of CASP1, CASP8, and CASP9 to CASP3/7 activation in THP-1 cells. In *CASP1, CASP8, and CASP9* single KO THP-1 cells, LPS transfection still induced robust cleavage of CASP3, CASP7, and GSDME, with no change in Sytox Green uptake (S2A-S2I Fig). CASP3/7 activity was unchanged in *CASP8* and *CASP9* KO cells but was significantly increased in CASP1 KO cells compared to WT cells (S2C, S2F, and S2I Fig). In *CASP1* KO cells, GSDMD processing was nearly abolished suggesting most GSDMD cleavage downstream of non-canonical inflammasome activation is mediated by CASP1 (S2A Fig). This is consistent with our previous findings that GSDMD is cleaved less efficiently in cells by CASP4 and CASP5 than by CASP1 [21].

Because CASP1, CASP8 and CASP3 can all cleave Bid, a Bcl-2 family protein that activates CASP9 to promotes CASP3/7 activation [31,32,49,50], we tested whether CASP4/5 activate CASP3/7 indirectly through cleavage of Bid. PMA-differentiated THP-1 lysates were incubated with recombinant caspases, and substrate cleavage was assessed at early time points before any secondary CASP3/7 activation. CASP1 and CASP8 cleaved Bid into its truncated form, as expected (Fig 2D). In contrast, CASP4 and CASP5 did not cleave Bid, although they were active and cleaved IL-18 and

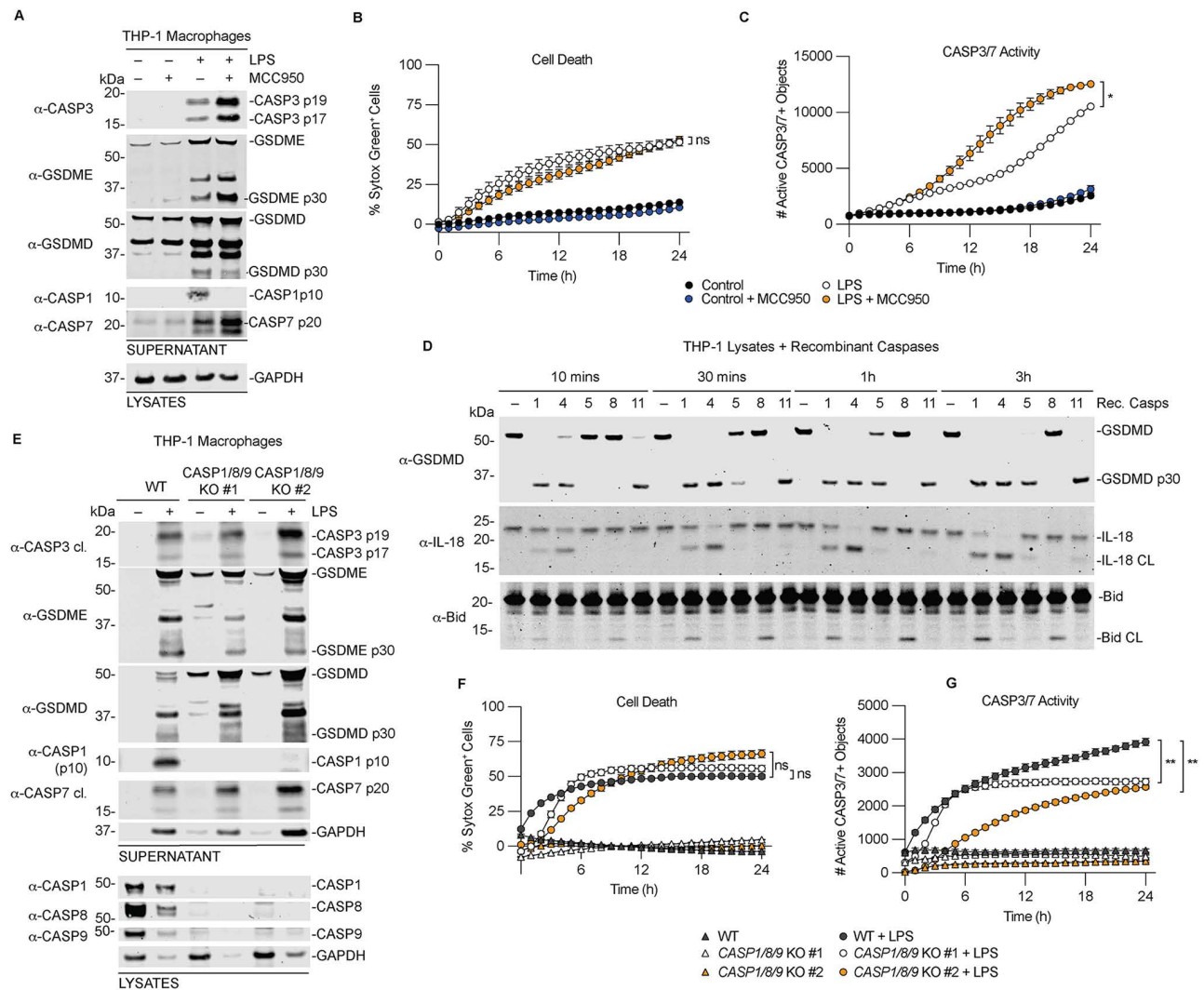

**Fig 2. CASP1, CASP8, and CASP9 are not required for CASP3/7 and GSDME processing during non-canonical inflammasome activation.**
**(A-C)** WT THP-1 cells were transfected with 25 µg/mL LPS in the presence or absence of the NLRP3 inhibitor MCC950 (10 µM) for 24 h. Supernatants and lysates were analyzed by immunoblotting **(A)**, cell death was measured by Sytox Green uptake **(B)**, and CASP3/7 activity was quantified using the CellTreat CASP3/7 detection reagent **(C)**. **(D)** PMA-differentiated WT THP-1 macrophages were lysed and incubated with 0.25 activity units/µL of recombinant caspases for the indicated times, followed by immunoblot analysis. **(E-G)** WT or *CASP1/8/9* triple KO cells were transfected with 25 µg/mL LPS for 24 h. Supernatants and lysates were analyzed by immunoblotting **(E)**, cell death was measured by Sytox Green uptake **(F)**, and CASP3/7 activity was measured using the CellTreat CASP3/7 detection reagent **(G)**. Data are mean±SEM of three independent replicates, and representative of at least three independent experiments except for D (n=2). ****$P < 0.0001$, ***$P < 0.001$, **$P < 0.01$, and *$P < 0.05$ by two-way ANOVA test with Tukey's multiple comparison test comparing the WT to KO treated samples at 24 h.

GSDMD (Fig 2D). In this assay, IL-18 processing was favored by CASP4 whereas GSDMD processing was favored by CASP1 (Fig 2D).

Finally, to rule out compensation among apoptotic caspases, we generated *CASP1/8/9* triple KO THP-1 cells. Similar to the single KO results, LPS transfection induced robust CASP3/7 processing with no change in Sytox Green uptake and only a minimal reduction in CASP3/7 activity (Fig 2D-2F). We also observed a decrease in full-length CASP8 and CASP9 in WT LPS-treated cells, suggesting activation. Because these caspases share specificity for the DEVD tetrapeptide sequence [51], which

forms the basis for the CASP3/7 reporter dye, they may contribute modestly to the measured activity. Collectively, these results demonstrate that CASP1, CASP8, and CASP9 are dispensable for CASP3/7 activation during noncanonical inflammasome signaling and support a model in which CASP4/5 directly cleave CASP3/7 to promote CASP3-dependent GSDME processing.

**CASP3 and CASP4/5 are required to induce GSDME processing during non-canonical inflammasome activation**

Our findings above suggest that CASP4/5 activate the executioner caspases CASP3/7, leading to CASP3-dependent GSDME cleavage. Because non-canonical inflammasomes detect intracellular LPS from Gram-negative bacteria, we next tested whether CASP4/5 cleave and activate CASP3/7 during LPS transfection and bacterial infection. WT, *CASP4/5* KO, and *CASP3* KO THP-1 cells were transfected with LPS or infected with *Stm* (Fig 3). CASP3/7 and GSDME processing were assessed by immunoblotting, cell death by Sytox Green uptake, and CASP3/7 activity using a cell-permeable reporter. We also examined the contribution of CASP7 to GSDME processing (S2 Fig).

Loss of CASP4/5 significantly impaired CASP3/7 activation, GSDME cleavage, Sytox Green uptake, and CASP3/7 activity following LPS transfection (Fig 3A-3C). In contrast, *CASP3* KO cells lacked GSDME cleavage but showed increased GSDMD processing (Fig 3D). Despite loss of GSDME cleavage, cell death was not significantly reduced (Fig 3E), suggesting compensation through enhanced GSDMD activation. CASP3 deficiency caused a modest reduction in overall CASP3/7 activity, likely reflecting preserved CASP7 activity (Fig 3F). *CASP7* KO cells retained CASP3 activation and GSDME processing, although at reduced levels compared to WT cells. Surprisingly, CASP1 processing into the p10 species and GSDMD cleavage were markedly reduced, resulting in delayed pyroptosis (S2J and S2K Fig). *CASP7* KO cells also showed a modest decrease in CASP3/7 activity, as expected (S2L Fig).

We tested whether CASP4/5 cleave and activate CASP3/7 during bacterial infection. WT, *CASP4/5* KO, and *CASP3* KO cells were infected with *Stm,* and CASP3/7 and GSDME processing, cell death, and CASP3/7 activity were assessed. Log phase *Salmonella* grown in SPI-1 inducing conditions express the SPI-1 type III secretion system, which activates both canonical and non-canonical inflammasome pathways [52–55]. Thus, we first used stationary phase bacteria (*Stm*-Stat.) to minimize the contribution of canonical CASP1 signaling (S3 Fig). Cells were infected across a range of multiplicities of infection (MOI) to model different infection burdens. In WT cells, we observed dose-dependent cleavage of GSDME and GSDMD (S3A and S3C Fig). CASP3/7 activation and GSDME processing were largely abolished in *CASP4/5* KO cells, whereas GSDMD processing was retained, likely due to activation of NLRC4 and or NLRP3 inflammasomes (S3A Fig) [55–57]. Similarly, GSDME, but not GSDMD, processing was lost in *CASP3* KO cells (S3C Fig). Despite cleavage of pore-forming proteins, little to no cell death was detected by Sytox Green uptake (S3B and S3D Fig), indicating that CASP4/5-dependent CASP3/7 activation and GSDME processing are insufficient to induce lysis under these conditions.

Because stationary-phase infection induced minimal cell death, and SPI-1 expression is required for efficient invasion and replication [15,58], we next focused on SPI-1-induced log-phase *Salmonella* (*Stm*^SPI-1). To further define the contribution of canonical inflammasome signaling, we analyzed WT, *CASP1* KO, and *NLRC4* KO cells [56] treated with the NLRP3 inhibitor MCC950 (S4 Fig). Compared to WT cells, *CASP1* KO cells showed reduced GSDMD and IL-1β processing but increased CASP3/7 activation and GSDME cleavage (S4A Fig). Consistent with this, cell death was unchanged, while CASP3/7 activity was significantly increased across all MOIs (S4B and S4C Fig). Increased PARP cleavage in *CASP1* KO cells further indicated activation of apoptotic pathways. *NLRC4* KO cells phenocopied *CASP1* KO cells, showing increased CASP3/7 activation and GSDME processing (S4D Fig). Consistent with non-canonical inflammasome signaling, *NLRC4* KO cells displayed increased Sytox Green uptake and CASP3/7 activity compared to WT cells, and this effect was not inhibited by MCC950 (S4E and S4F Fig). *NLRC4* KO cells also exhibited increased GSDMD processing, which was reduced by MCC950. Notably, GSDMD and IL-1β processing in both WT and *NLRC4* KO cells were inhibited by MCC950 pretreatment, indicating that NLRP3 is the primary mediator of CASP1 activation and substrate cleavage in this context. Together, these data indicate that canonical CASP1 signaling downstream of NLRP3 or NLRC4 is required for optimal GSDMD and IL-1β processing but is not required for CASP3/7 activation or GSDME cleavage.

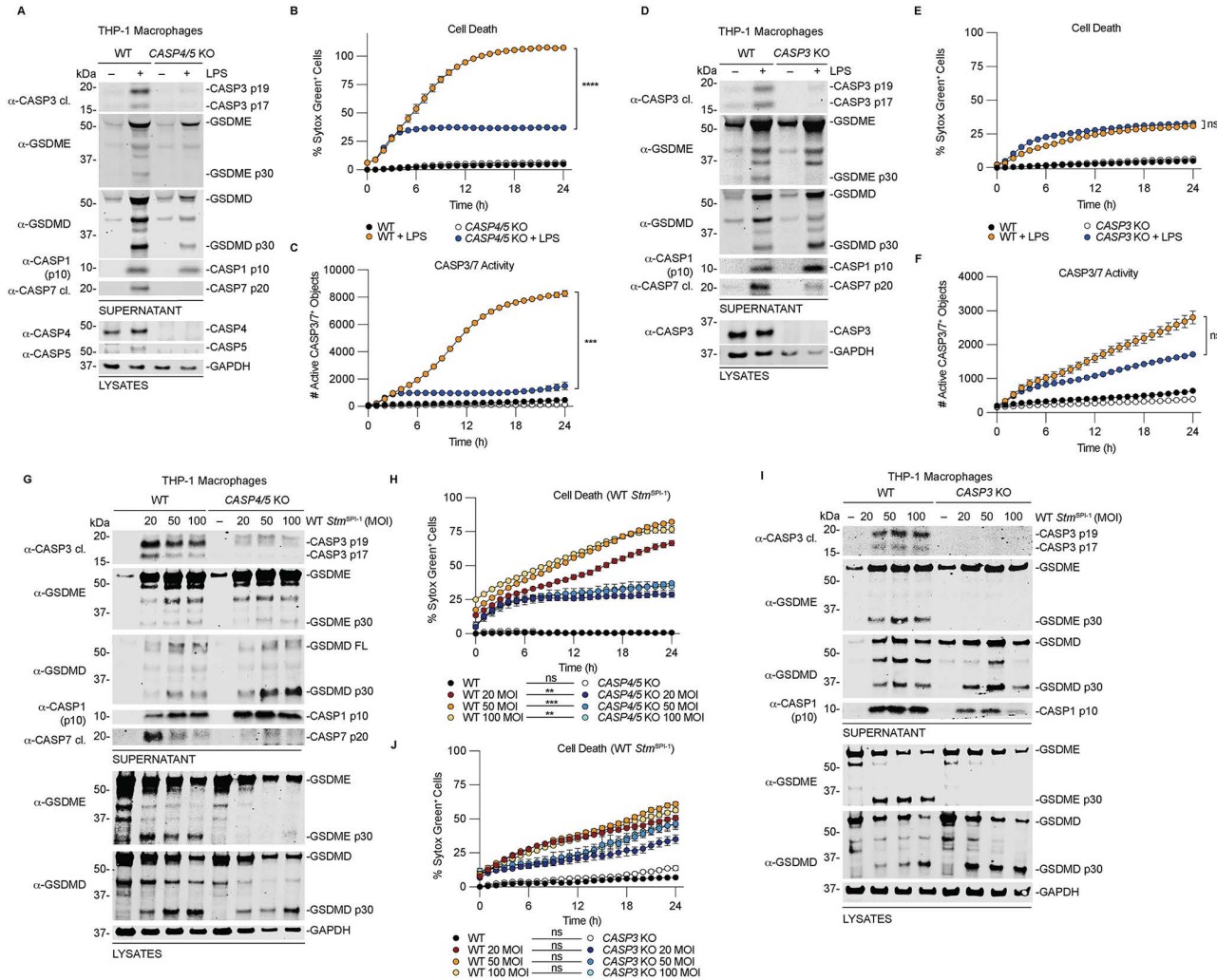

**Fig 3. CASP3 and CASP4/5 are required to induce GSDME processing during non-canonical inflammasome activation. (A-F)** WT and *CASP4/5* double KO **(A-C)** or WT and *CASP3* KO **(D-F)** THP-1 cells were transfected with 25 µg/mL of LPS for 24 h. Supernatants and lysates were analyzed by immunoblotting **(A,D)**, cell death was measured by Sytox Green uptake **(B,E)**, and CASP3/7 activity was determined using CellTreat CASP3/7 detection reagent **(C,F)**. **(G-J)** WT and *CASP4/5* KO **(G,H)** or WT and *CASP3* KO THP-1 macrophages **(I,J)** were infected with WT *Stm*<sup>SPI-1</sup> at the indicated MOI for 24 h. Supernatants and lysates were analyzed by immunoblotting **(G,I)**, and cell death was measured by Sytox Green uptake **(H,J)**. Data are mean ± SEM of three independent replicates, and representative of at least three independent experiments. ****P < 0.0001, ***P < 0.001, **P < 0.01, and *P < 0.05 by two-way ANOVA test with Tukey's multiple comparison test comparing the WT to KO treated samples at 24 h.

We next directly tested the role of CASP4/5 in activating CASP3/7 and GSDME during infection. WT, *CASP4/5* KO and *CASP3* KO cells were infected with *Stm*<sup>SPI-1</sup> (Fig 3G-3J). As observed with LPS transfection, CASP4/5 were required for CASP3/7 activation, and CASP3 was required for GSDME processing (Fig 3G and 3I). Although CASP1 was activated and could contribute to CASP3/7 activation, loss of CASP4/5 markedly impaired CASP3/7 activation. Functionally, CASP4/5 deficiency reduced Sytox Green uptake, whereas CASP3 deficiency did not (Fig 3H and 3J). Together, these findings demonstrate that CASP4/5 activate CASP3/7 during non-canonical inflammasome activation, leading to CASP3-dependent GSDME cleavage. However, CASP3 is dispensable for lytic cell death, likely due to compensatory GSDMD activation. Of note, CASP3 can cleave and inactivate GSDMD and inhibiting this inactivation enhances

pyroptosis [29,59]. Similarly, loss of CASP1 increases GSDME processing without reducing cell death, highlighting the plasticity of pyroptotic execution pathways.

**GSDMD is the primary mediator of LPS-induced cell death but is dispensable for *Salmonella*-induced cell death**

Mammals encode six gasdermins: gasdermin A, B, C, D, E, and PJVK (also known as DFNB59) [38,60,61]. Their N-terminal pore-forming domains are typically autoinhibited by their C-termini. Upon cleavage, the N-terminal fragments are released and form membrane pores that induce pyroptosis. Inflammatory caspases cleave GSDMD to induce pyroptosis [17–19]. CASP3 can cleave GSDME to induce pyroptosis, but this has been mainly described in the context of chemotherapy-induced apoptosis or cytotoxic lymphocyte-mediated cell death [40,62,63]. Whether CASP3/GSDME contribute to pyroptosis during non-canonical inflammasome activation remains unclear.

To address this, we transfected WT, *GSDMD* KO, *GSDME* KO, or *GSDMD/E* double KO THP-1 macrophages with LPS and assessed gasdermin processing, cell death, CASP3/7 activation, and IL-1α/β release (Fig 4A-4E). Immunoblot analysis revealed increased CASP3/7 processing and reduced CASP1 processing in all KO cells compared to WT cells following LPS transfection (Fig 4A). This suggests both GSDMD and GSDME may contribute to downstream CASP1 activation through the NLRP3 inflammasome. Loss of either GSDMD or GSDME increased CASP3 and CASP7 activation compared to WT cells, although CASP3/7 activity was significantly elevated only in *GSDME* KO cells (Fig 4D). GSDME processing was increased in *GSDMD* KO cells, but GSDMD processing was attenuated in *GSDME* KO cells compared to WT cells (Fig 3A). Remarkably, GSDME loss did not affect Sytox Green uptake or LDH release whereas GSDMD deficiency markedly reduced both readouts (Fig 4B and 4C). *GSDMD/E* double KO cells phenocopied *GSDMD* KO cells (Fig 4B and 4C), indicating that GSDMD is the primary mediator of lytic cell death during intracellular LPS sensing.

In contrast, IL-1α/β release required both gasdermins, as measured by the HEK Blue assay which reports on the activity of secreted active IL-1α/β [21] (Fig 4E). Loss of either GSDMD or GSDME alone did not significantly reduce cytokine release, but combined loss of both resulted in a marked decrease, consistent with their cooperative roles in cytokine secretion [46]. These data indicate that GSDMD is the dominant driver of pyroptotic cell death, whereas both GSDMD and GSDME contribute to cytokine release during non-canonical inflammasome activation.

We next examined the contribution of gasdermins during infection with *Stm*^SPI-1 (Fig 4F-4J). CASP3/7 activation patterns were similar to those observed during LPS transfection, but key differences were observed in cell death. GSDME processing increased in *GSDMD* KO cells, and GSDMD processing increased in *GSDME* KO cells. Unlike LPS treatment, GSDMD processing was not reduced in *GSDME* KO cells, likely due to increased CASP1 activation during *Stm*^SPI-1 infection (Fig 4F). Indeed, CASP1 processing was comparable between WT and *GSDME* KO cells but reduced in *GSDMD* and *GSDMD/E* KO cells. Unexpectedly, cell death, measured by Sytox Green uptake and LDH release, was not reduced in any of the knockout lines at 24 hours post infection and in some cases was increased (Fig 4G-4J). These findings suggest that additional pore-forming mechanisms contribute to cell lysis during infection.

Consistent with this idea, *GSDMD/E* double KO cells displayed lytic morphology indicative of gasdermin-independent membrane rupture (S5A Fig). This suggests that increased CASP3/7 activity may drive cell lysis through alternative pathways such as Ninjurin-1 (NINJ1)-mediated membrane rupture [15,64] or pannexin-1 (PANX1), a known CASP3/7 substrate capable of forming membrane pores independently of gasdermins [65]. To test this, WT, *GSDMD* KO and *GSDMD/E* KO cells were infected with *Stm*^SPI-1 at a MOI of 50 following pretreatment with spironolactone, a PANX1 inhibitor, glycine, an osmoprotectant that inhibits NINJ1-mediated lysis, or both [64,65]. Inhibition of PANX1 or NINJ1 alone had modest effects, whereas combined inhibition significantly reduced LDH release across all genotypes (S5B Fig). These findings support a model in which PANX1 and NINJ1 act downstream of apoptotic signaling to promote infection-induced cell lysis [66]. In agreement with this, increased PANX1 processing was observed in cells with elevated CASP3/7 activation following infection, including *CASP1*, *NLRC4*, *GSDMD*, *GSDME*, and *GSDMD/E* KO cells (S4 and 4F Figs). Combined inhibition of PANX1 and NINJ1 did not affect pore formation in WT cells, as measured by Sytox Green uptake, but

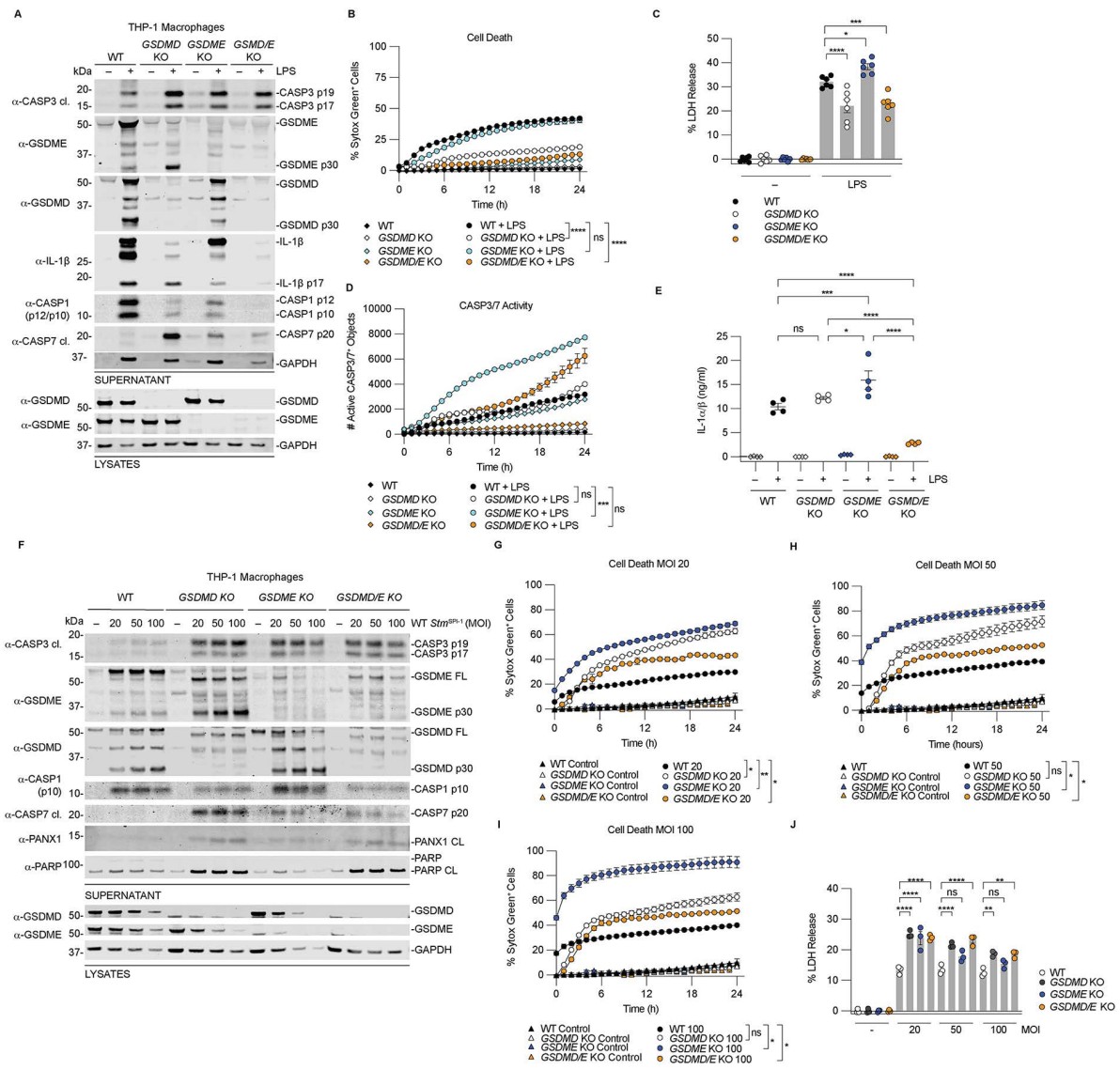

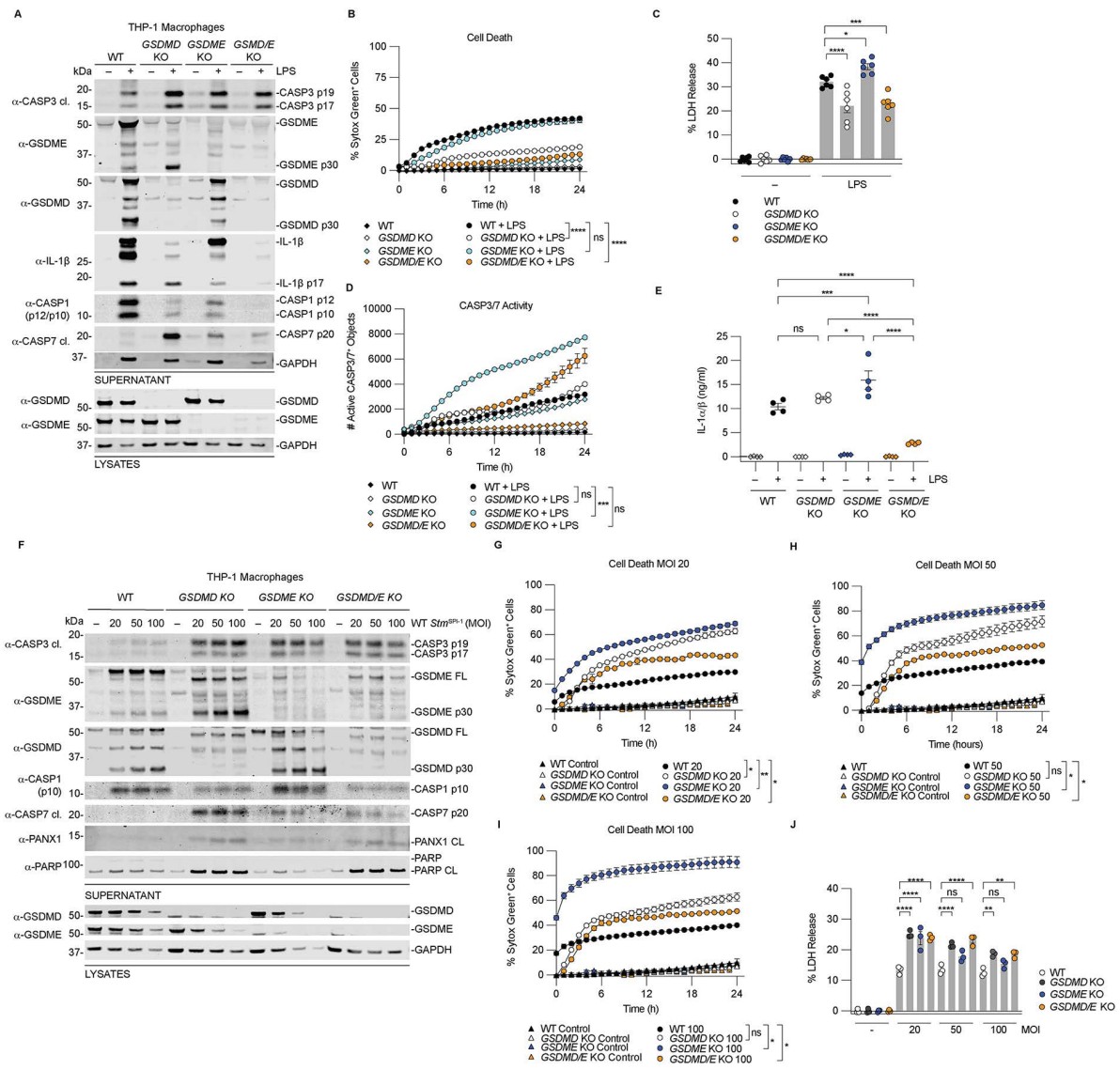

**Fig 4. GSDMD is the primary mediator of LPS-induced cell death but is dispensable for *Salmonella*-induced cell death. (A-E)** WT, *GSDMD* KO, *GSDME* KO or *GSDMD/E* KO THP-1 macrophages were transfected with 25 µg/mL of LPS for 24 h then supernatants and lysates were analyzed by immunoblotting **(A)**, cell death was measured by Sytox Green uptake (**B**) and LDH release **(C)**, CASP3/7 activity was determined using CellTreat CASP3/7 detection reagent **(D)**, and active IL-α/β released into the supernatants was quantified **(E)**. **(F-J)** WT, *GSDMD* KO, *GSDME* KO and *GSDMD/E* KO THP-1 macrophages were infected with WT *Stm*<sup>SPI-1</sup> at the indicated MOI of for 24 h. Supernatants and lysates were analyzed by immunoblotting **(F)**, and cell death was measured by Sytox Green uptake (**G-I**) and LDH Release **(J)**. Data are mean±SEM of three independent replicates, and representative of at least three independent experiments except C (n=2). ****P<0.0001, ***P<0.001, **P<0.01, and *P<0.05 by one-way ANOVA test with Tukey's multiple comparisons test for C, E, J or two-way ANOVA test with Tukey's multiple comparison test comparing the WT to KO treated samples at 24 h.

significantly reduced Sytox Green uptake in *GSDMD/E* KO cells (S5C and S5D Fig). PANX1 inhibition alone had little effect, whereas NINJ1 inhibition reduced Sytox Green uptake, suggesting that NINJ1 is the primary mediator of membrane rupture under these conditions. Collectively, these findings indicate that GSDMD is the dominant pore-forming protein driving cell lysis during intracellular LPS sensing, whereas both GSDMD and GSDME contribute to cytokine release.

In contrast, during *Salmonella* infection, gasdermin-independent mechanisms, including PANX1 and NINJ1, can drive cell rupture downstream of CASP3/7 activation.

## GSDMD and CASP3 are required, but GSDME is dispensable for restricting *Salmonella* replication in human macrophages

Recent work has shown that CASP4 limits intracellular *Salmonella* replication, particularly at later stages of infection, although the underlying mechanism remains unclear [15]. Pyroptosis is generally considered host-protective because it eliminates intracellular replicative niches and promotes cytokine release that enhances immune responses. We therefore hypothesized that CASP4 restricts replication by activating CASP3 to induce GSDME-mediated pyroptosis when bacterial burden increases.

To test the role of GSDME in restricting *Stm* replication, we infected WT, *GSDMD*, *GSDME,* and *GSDMD/E* KO THP-1 macrophages with GFP-expressing *Stm (GFP-Stm)* at increasing MOIs (MOI 20, 50, and 100). Bacterial replication was quantified by measuring changes in GFP intensity over time (Figs 5 and S6). Imaging was initiated 2 hours post infection, and GFP-*Stm* replication kinetics were monitored for 24 hours using an Incucyte system. The fold change in GFP intensity at 24 hours was calculated relative to the starting time point, which corresponds to 2 hours post infection (S6A, S6C, and S6D Fig). Unexpectedly, GSDME was dispensable for limiting bacterial replication (S6A–S6D Fig). Only *GSDMD* KO cells exhibited significantly increased bacterial burden compared to WT cells, consistent with previous reports demonstrating that GSDMD is required to restrict intracellular *Salmonella* replication [15]. In contrast, *GSDME* KO cells showed reduced bacterial burden compared to WT cells, suggesting that GSDME may promote infection or replication. Consistent with this, *GSDMD/E* double KO cells displayed bacterial burdens similar to WT cells and lower than *GSDMD* KO cells. Cell death, measured by LDH release, was comparable across all genotypes (S6B Fig), indicating that differences in bacterial replication are not explained by differences in cell death.

We next examined the role of CASP3 in restricting bacterial replication. Because *CASP7* KO cells exhibited defects in GSDMD processing (S2J Fig), we focused on CASP3. WT and *CASP3* KO cells were infected with GFP-*Stm* and analyzed by immunoblotting, LDH release, and live-cell imaging (Figs 5 and S6). *CASP3* KO cells showed similar levels of GSDMD processing and cell death as WT cells (Fig 5A and 5B), indicating that CASP3 is not required for lytic cell death. Interestingly, *CASP3* KO cells exhibited reduced CASP7 activation (Fig 5A), suggesting CASP7 activation could partially depend on CASP3 in this context.

Despite similar levels of cell death, *CASP3* KO cells showed a significant increase in bacterial burden across all MOIs compared to WT cells (Figs 5C, S6E, and S6F). To more precisely quantify replication, we performed high-resolution imaging and counted the number of bacteria per cell (Fig 5D and 5E). *CASP3 KO* cells contained more bacteria per cell at 24 hours compared to WT cells, indicating a defect in restricting bacterial replication. Importantly, the number of bacteria per cell at the start of infection was similar between genotypes, indicating that initial infection levels were comparable. Together, these results demonstrate that GSDMD and CASP3 are required to restrict intracellular *Salmonella* replication in human macrophages, whereas GSDME is dispensable.

## CASP3/7 and GSDME are cleaved in primary human macrophages during non-canonical inflammasome activation

To evaluate the physiological relevance of CASP4/5-mediated CASP3 and GSDME activation, we transfected primary human monocyte-derived macrophages, differentiated with macrophage colony stimulating factor, with LPS [67]. Consistent with results in THP-1 cells, LPS transfection induced robust processing of CASP3/7 and GSDME in primary macrophages (Fig 6A). Representative immunoblots from three independent donors are shown. Pretreatment with the NLRP3 inhibitor MCC950 blocked CASP1 activation and reduced GSDMD processing but did not affect CASP3/7 or GSDME cleavage. These findings indicate that CASP3/7 activation is primarily mediated by CASP4/5, whereas GSDMD processing occurs largely downstream of CASP1 (Fig 6A).

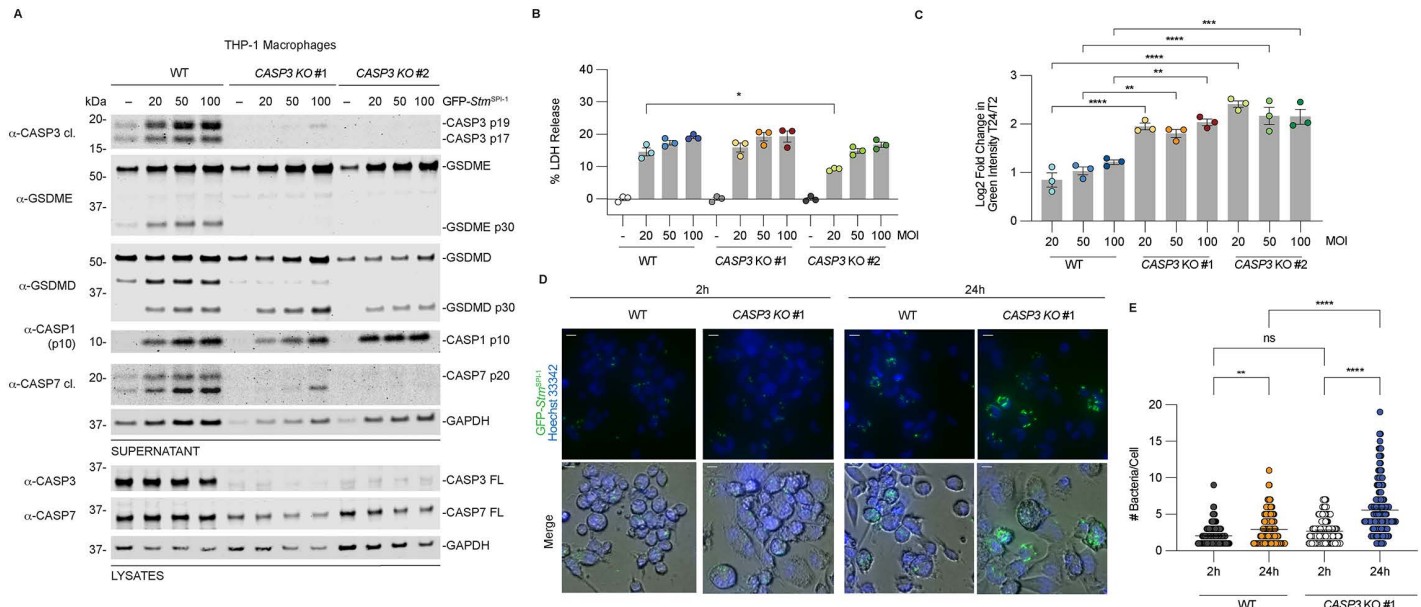

**Fig 5. CASP3 limits *Salmonella* replication in human macrophages. (A,B)** WT or *CASP3* KO THP-1 macrophages were infected with GFP-*Stm*^SPI-1 at the indicated MOI for 24 h. Supernatants and lysates were analyzed by immunoblotting (**A**) and LDH release (**B**). **(C)** WT or *CASP3* KO THP-1 macrophages were treated as in A and the $log_2$ fold change in green intensity was quantified as a measure of bacterial replication. **(D,E)** WT or *CASP3* KO THP-1 macrophages were infected with GFP-*Stm*^SPI-1 at MOI = 20. Images were collected (**D**) and the number of bacteria per cell was quantified from an equivalent number of cells across triplicate wells per genotype **(E)**. Scale bars in D indicate 10 μm. Data are mean ± SEM of three independent replicates, and representative of at least three independent experiments. ****$P < 0.0001$, ***$P < 0.001$, **$P < 0.01$, and *$P < 0.05$ by one-way ANOVA test with Tukey's multiple comparisons test for B, C and two-way ANOVA test with Tukey's multiple comparison test in E.

Cell death and CASP3/7 activity induced by intracellular LPS were comparable between untreated and MCC950-treated cells (Fig 6B-6D), consistent with non-canonical inflammasome activation. Representative images from Donor 7 show pyroptotic morphology and Sytox Green uptake used to monitor cell death kinetics (Fig 6B). Quantification across all donors confirmed that cell death was not altered by MCC950 treatment (Fig 6E). Importantly, *Salmonella* infection also induced processing of CASP3/7, GSDME, and GSDMD in primary macrophages (Fig 6F). Together, these results demonstrate that intracellular LPS and *Salmonella* primarily engage CASP4/5 to drive CASP3 activation and GSDME processing in primary human cells.

## Discussion

Cell death is essential for maintaining cellular homeostasis and controlling infection, with inflammasome-mediated pyroptosis serving as a critical defense mechanism against intracellular pathogens [68]. Crosstalk between apoptotic and pyroptotic caspases is well established in canonical inflammasome signaling, where CASP1 can activate the executioner caspases CASP3 and CASP7 [23,29,31,32,69–72]. In human cells, CASP1 can directly cleave CASP3/7 under certain conditions, such as DPP8/9 inhibition [29], whereas in mouse macrophages CASP1-dependent activation of CASP3/7 typically requires CASP8/9 via Bid cleavage in response to diverse inflammasome agonist [31,32]. Whether human non-canonical inflammasomes (CASP4/5) similarly activate apoptotic executioner caspases has remained unclear. Here, we define a direct CASP4/5-CASP3 signaling axis that links non-canonical inflammasome activation to apoptotic executioner caspases. Unlike CASP1, CASP4/5 do not cleave Bid, and CASP1, CASP8 and CASP9 are dispensable for CASP3/7 activation.

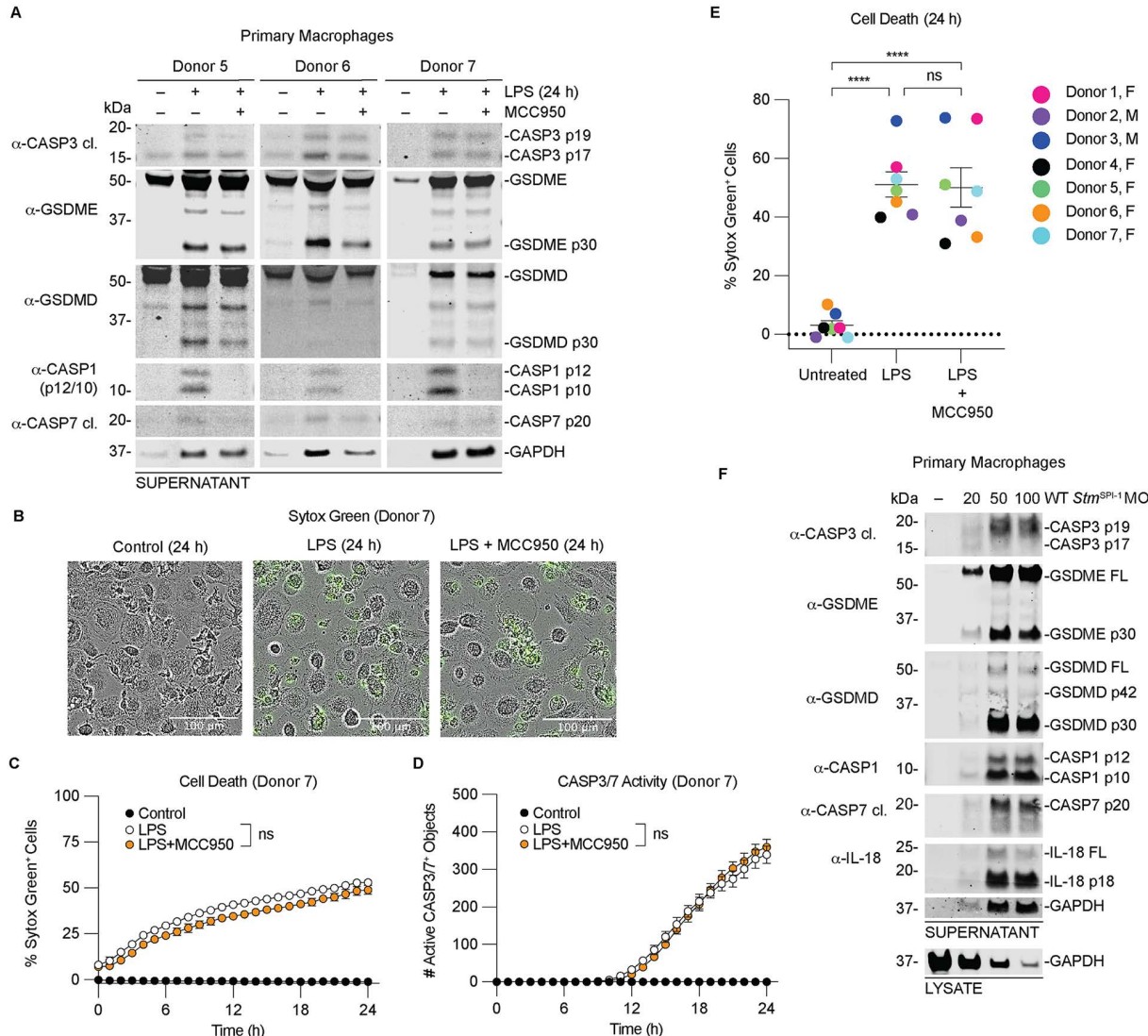

**Fig 6. CASP3/7 and GSDME are cleaved in primary human macrophages during non-canonical inflammasome activation. (A-F)** Primary human monocytes were differentiated into macrophages and transfected with 25 µg/mL of LPS for 24 h. Where indicated, cells were pretreated with 10 µM MCC950 for 30 minutes before LPS transfection. Supernatants were analyzed by immunoblotting **(A)**, cell death was assessed by Sytox Green uptake for a representative donor (Donor 7) **(B,C)**, and CASP3/7 activity measured using the CellTreat detection reagent **(D)**. Quantification of Sytox Green uptake across all donors is shown **(E)**. **(F)** Differentiated primary macrophages were infected with the indicated MOIs of WT $Stm^{SPI-1}$ for 24 h before immunoblotting. Data are mean±SEM of three or more independent experiments ****P<0.0001, ***P<0.001, **P<0.01, and *P<0.05 by two-way ANOVA test with Tukey's multiple comparison test comparing the control vs treated samples at 24 h. Panel E was analyzed by One-way Anova with Tukey's multiple comparisons test.

Our findings reveal reciprocal regulation between the CASP1/GSDMD axis and the CASP4/5-mediate CASP3/GSDME pathway. Loss of CASP1 reduced GSDMD processing but enhanced CASP3/7 activation and GSDME cleavage, suggesting that CASP4/5 preferentially drive apoptotic caspases activation when CASP1 activity is limited. This likely reflects the faster kinetics of GSDMD-mediated pyroptosis compared to apoptotic caspase activation [29,73]. Although GSDMD has long been recognized as the primary effector of LPS-induced pyroptosis [17–19], our data suggest that in human macrophages, most GSDMD processing is mediated by CASP1. In contrast, CASP4/5 preferentially promote CASP3-dependent

GSDME activation, which contributes to downstream NLRP3 activation. Given that CASP4/5 and CASP3 can inactivate key inflammatory substrates, including IL-1β, IL-18, and GSDMD [21,29,74–76], this pathway may allow cell death while limiting excessive inflammation.

Strikingly, we observed residual GSDMD processing in CASP4/5-deficient cells suggesting the presence of alternative LPS-sensing pathways. One possibility is NLRP11, a human-specific sensor that can activate the NLRP3 inflammasome in response to LPS [77,78]. This redundancy is likely important during infection, as pathogens frequently target and inactivate pyroptotic machinery. For example, SARS-CoV-2 NSP5 and enterovirus 3C proteases cleave and inactivate GSDMD, while *Shigella flexneri* promotes its degradation, promoting infection [79,80]. In these contexts, compensatory activation of CASP3/GSDME may preserve cell death responses. Consistent with this, we observed increased CASP3/7 activation and GSDME cleavage in GSDMD-deficient cells. Thus, redundant pathways may ensure robust cell death and cytokine release when one arm is inhibited.

Despite robust GSDME activation, our data indicate that GSDME-mediated pores are largely sublytic and do not drive terminal lysis. Instead, GSDMD remains the dominant mediator of lytic cell death following LPS stimulation, consistent with recent findings using mouse macrophages [66]. However, during *Salmonella* infection, neither GSDMD nor GSDME was required for cell lysis at 24 h, in contrast to earlier work implicating GSDMD and NINJ1 at earlier time points [15]. These differences may reflect variations in infection kinetics (6 h vs 24 h in this study), priming conditions (Pam3CSK4 vs. unprimed in our study), or engagement of alternative pathways. Emerging evidence supports a role for PANX1 channels and NINJ1-mediated membrane rupture downstream of apoptotic caspases [59,65,66]. Indeed, NINJ1-dependent lysis can occur independently of the typical pore-forming proteins, including gasdermins and MLKL (mixed lineage kinase domain-like protein), under specific conditions [64,81]. Together, these findings suggest that multiple, partially redundant mechanisms can drive NINJ1 unzipping and membrane rupture downstream of inflammasome and apoptotic signaling [82].

Although GSDME does not contribute substantially to lytic cell death, both GSDMD and GSDME promote cytokine release. While single deletion of either gasdermin had minimal impact on IL-1 activity, combined loss reduced IL-1 release, consistent with cooperative roles in cytokine secretion [46]. These findings further support a model in which distinct pore-forming proteins contribute differentially to lysis and cytokine release.

Controlled activation of pyroptosis typically protects the host from infection. Therefore, the finding that GSDME is dispensable for controlling *Salmonella* replication raises important questions about the role of sublytic pores during infection. The reason GSDME fails to restrict bacterial replication remains unclear. One possibility is that sublytic pores prolong host cell survival, allowing pathogens additional time to replicate before eventual cell lysis. In addition, this may reflect the absence of a full CASP1- and GSDMD-driven inflammatory response, which is likely critical for effective pathogen restriction. Supporting this idea, CASP11-mediated cell death in the absence of CASP1 increases susceptibility to *Salmonella* infection [52]. Alternatively, GSDMD may possess stronger direct antimicrobial activity than GSDME [83,84]. Consistent with this notion, GSDMD, but not any other gasdermins, has been shown to protect against *Salmonella* infection in the gut of mice [85]. The contribution of GSDME likely depends on the pathogen and cellular context, as it can be protective in viral infections [37].

A key question is how executioner caspases mechanistically restrict infection. Although apoptosis has traditionally been viewed as immunologically silent, growing evidence indicates that it contributes to host defense [33,35,66,86–88]. Our data support this paradigm by showing that CASP4/5-driven activation of CASP3 limits *Salmonella* replication. Notably, because LDH release was unchanged in *CASP3* KO cells, these findings suggest that executioner caspases restrict bacterial replication through mechanisms independent of lytic cell death.

In addition to apoptosis, a plausible mechanism is the cleavage of host or bacterial proteins. For example, CASP3 and CASP7 can cleave host substrates such as PANX1, which has been shown to be important for inhibiting *Salmonella* replication [66]. CASP3 has also been reported to cleave effector proteins from *Listeria* and *Salmonella*, with functional

consequences for bacterial replication [86,89]. To evade this host defense mechanism, some pathogens, such as adherent invasive *Escherichia coli*, promote CASP3 proteasomal degradation, enabling bacterial persistence [90].

We also observed reduced CASP7 activation in *CASP3* knockout cells, suggesting that loss of both activities may contribute to the defect in restricting replication. This may indicate that CASP3 contributes to CASP7 activation in this context. Alternatively, loss of CASP3 activity may prevent the processing and inactivation of bacterial effectors that inhibit CASP7 activation. Consistent with this idea, CASP7 deficiency enhanced bacterial growth in THP-1 cells, whereas CASP3 and CASP7 exhibit context-dependent roles across pathogens and cell types [33]. *In vivo* studies further support protective roles for executioner caspases, including CASP7 in controlling *Legionella* and CASP3/7 in protection against *Clostridium difficile* infection independently of the Pyrin inflammasome [87,88,91].

In summary, we identify CASP3 and CASP7 as direct substrates of the human non-canonical inflammasomes CASP4 and CASP5, defining a pathway that links inflammatory sensing to activation of apoptotic executioner caspases (Fig 7). Although CASP3 activation leads to GSDME cleavage, our data show that CASP3 and GSDMD, but not GSDME, are required to restrict intracellular *Salmonella* replication (Fig 7). Complementary biochemical studies further support direct recognition of CASP3 and CASP7 by CASP4 and CASP5, primarily through exosite interactions [92]. Together, these findings reveal a previously unrecognized role for executioner caspases in innate immunity and refine our understanding of how non-canonical inflammasomes coordinate cell death and pathogen control. Future studies should define how CASP4/5 and CASP3 regulate pathogen replication in primary cells and identify the host and bacterial substrates that mediate this protective effect.

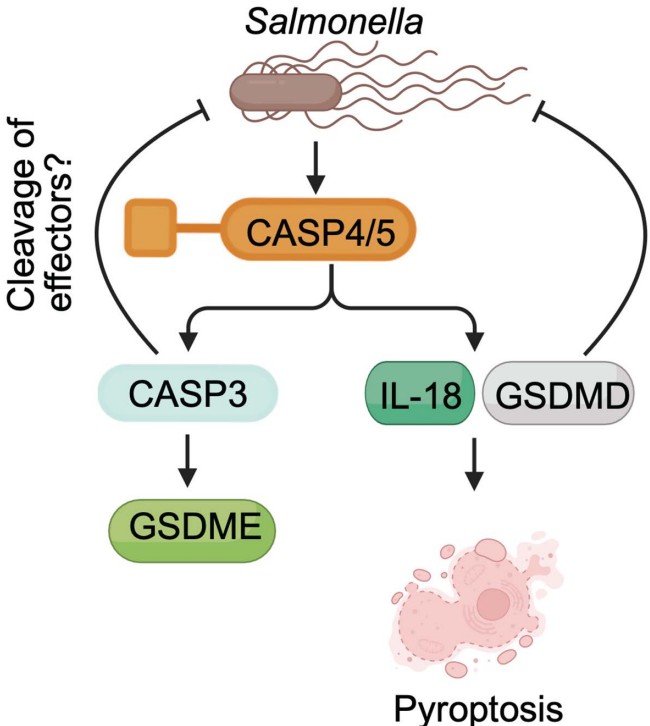

**Fig 7. Proposed model of non-canonical inflammasome signaling that limits intracellular *Salmonella* replication.** Upon sensing intracellular LPS or *Salmonella*, CASP4/5 directly cleave CASP3/7. Activated CASP3 cleaves GSDME, but GSDME does not significantly contribute to cell lysis and is dispensable for controlling bacterial replication. In contrast, CASP3 activation is required to limit intracellular *Salmonella* replication. CASP4/5 also promote pyroptosis through cleavage of GSDMD and IL-18, and GSDMD is essential for restricting bacterial replication. Created in BioRender. Taabazuing, C. (2026) https://BioRender.com/59nsy1c.

## Methods

### Antibodies and reagents

Antibodies used include: GSDMD Rabbit polyclonal Ab (Novus Biologicals, NBP2–33422), FLAG M2 monoclonal Ab (Sigma, F3165), GAPDH Rabbit monoclonal Ab (Cell Signaling Tech, 14C10), CASP1 p12/10 Rabbit monoclonal Ab (Abcam, ab179515), CASP4 Rabbit polyclonal Ab (Cell Signaling Tech, 4450S), CASP5 Rabbit monoclonal Ab (Cell Signaling Tech, 46680S), Myc Mouse monoclonal Ab (Cell Signaling Tech, 2276S), CASP3 (Cell Signaling Tech 9664S), CASP3 (Cell Signaling Tech, 9662), CASP7 (Cell Signaling Tech, 8438S), CASP7 (Cell Signaling Tech, 12827S), GSDME (Abcam, ab215191), hIL-1β Goat Polyclonal Ab (R&D systems, AF-201-NA), hIL-18 Goat Ab (R&D systems, af2548), CASP8 (Cell Signaling Tech, 4790S), CASP9 (Cell Signaling Tech, 9508S), Bid (Cell Signaling Technology, 2002), IL-1α (Novus, AF-200-NA), HA Rabbit monoclonal Ab (Cell Signaling Tech, 3724S), FKBP12 Rabbit polyclonal Ab (Abcam, ab24373), Pannexin-1 (D9M1C) (Cell Signaling Tech, 91137), PARP (Cell Signaling Tech, 9542S), NLRC4 (D5Y8E) (Cell Signaling Tech, 12421S). IRDye 800CW anti-rabbit (LICORbio, 925–32211), IRDye 800CW anti-mouse (LICORbio, 925–32210), IRDye 680CW anti-rabbit (LICORbio, 925–68073), IRDye 680CW anti-mouse (LICORbio, 925–68072). Other reagents used include: LPS-EB Ultrapure (Invivogen, tlrl-3pelps), phorbol 12-myristate 12-acetate (PMA) (Promega, V1171), FuGENE HD (Promega, E2311), AP20187 (Tocris 62–975), MCC950 (Sigma Aldrich, 5381200001), Sytox Green (Invitrogen, S7020), CellEvent Caspase-3/7 Detection Reagent (Invitrogen, C10423).

### Recombinant caspase assays

All *in vitro* assays using recombinant caspases were performed in 20 μL reactions containing caspase assay buffer (20 mM PIPES, 100 mM NaCl, 10 mM DTT, 1mM EDTA, 0.1% CHAPS, 10% sucrose, pH 7.) as previously described [23]. 10 μg of plasmids encoding for GSDMD, CASP3, CASP7, or GSDME were transfected into HEK 293T cells (10 cm plates) using FuGENE HD according to the manufacturer's protocol. After 24 hours, cells were harvested and lysed by sonication for 10 second pulses for 60 seconds at 30% amplitude in PBS and centrifuged at 12,000 g to remove cell debris. Lysates were diluted 10x in caspase assay buffer and incubated with 0.25 activity units/μL of each indicated caspase. At the indicated time points, samples were mixed 1:1 with 2x Licor Protein Sample Loading Buffer (LICORbio, 928–40004) and boiled at 95 °C for 10 minutes, then analyzed by immunoblotting.

### CASP3/7 protein purification and cleavage assays

$3 \times 10^6$ HEK 293T cells were seeded in 10 cm culture plates in DMEM. The following day, the indicated plasmids were mixed in Opti-MEM and transfected using FuGENE HD according to the manufacturer's protocol. After 3 days, cells were washed 2x with cold PBS, harvested into 0.5 mL PBS and sonicated for 60 seconds with 10 second pulses. After sonication, samples were spun at 20,000g for 3 minutes and supernatants were mixed with 100 μL of washed Agarose Gel Anti-FLAG (Sigma-Aldrich, A2220) overnight. The following day, Anti-FLAG gel was washed 3x with cold PBS then incubated with 300 μg/mL 3x FLAG peptide (Sigma-Aldrich, F4799) for 1 hour at room temperature before eluting. Eluates were mixed with caspase buffer and cleavage assays were run according to *Recombinant Caspase Assays*.

### Cell culture

All cells were grown at 37 °C in a 5% $CO_2$ atmosphere incubator. HEK 293T cells were purchased from ATCC and THP-1 cells were obtained previously or acquired from the Bryant [21], Abbott [46], or Shin lab [56]. HEK 293T cells were cultured in Dulbecco's Modified Eagle's Medium (DMEM) with L-glutamine and 10% fetal bovine serum (FBS). THP-1 cells were cultured in Roswell Park Memorial Institute (RPMI) medium 1640 with L-glutamine and 10% fetal bovine serum (FBS). Purified human monocytes from healthy donors were purchased from the University of Pennsylvania Human Immunology Core. Purified monocytes were cultured in RPMI + 10% FBS and 25 ng/mL rm-csf in 96 well plates at a density of 100,000

cells/well. Media was removed and replaced with Opti-MEM containing Sytox Green and MCC950 as indicated for 30 minutes prior to LPS transfection as previously described. Cell lines were negative for mycoplasma contamination.

## IncuCyte CASP3/7 activity and Sytox Green uptake assays

The kinetics of CASP3/7 activity and Sytox Green uptake were measured in cells using the IncuCyte S3 Live cell imaging system (Sartorius). THP-1 cells were resuspended in RPMI medium containing 50 ng/mL phorbol 12-myristate 12-acetate (PMA) and plated in black, clear-bottom 96-well plates at a density of $8 \times 10^4$ cells/well. After 48 h, the media was then replaced with Opti-MEM (0.1 mL/well) supplemented with either CellEvent Caspase-3/7 Detection Reagent (final concentration: 10 µM (1:200 dilution)) or Sytox Green (final concentration: 0.2-0.5 µM), as indicated. A series of images were collected with a 10x objective at 1h intervals for 24 hours. The number of Sytox Green or CASP3/7 dye-positive cells in each image was determined using the IncuCyte S3 software with 3 different areas/well. Cell death was quantified relative to 0.1% Triton treated macrophages.

## Cloning

All plasmids were cloned using Gateway technology as previously described [21]. DNA encoding the indicated proteins were inserted between the *attR* recombination sites and shuttled into modified pLEX_307 vectors (Addgene) using Gateway technology (Thermo Fisher Scientific) according to the manufacturer's instructions. Proteins expressed from these modified vectors contain an N-terminal *att*B1 linker (GSTSLYKKAGFAT) after any N-terminal tag or a C-terminal *att*B2 linker (DPAFLYKVVDI) preceding any C-terminal tag such as Myc or HA. All plasmids were verified by DNA sequencing (Azenta).

## THP-1 knockout cell lines

sgRNAs were designed using the Broad Institute's web portal and cloned into the pXPR016_Hygro or plentiGuide-Puro vector (pXPR003) (Addgene #52963) as described previously [29]. The sgRNA sequences used were:

h*CASP3 clone#1* 5′-CGTGGTACAGAACTGGACTG -3′
h*CASP3* clone#2 5′- TGTCGATGCAGCAAACCTCA -3′
h*CASP7* 5′- AGACAATCACGTCAAAACCC -3′
h*CASP9* 5′- ACATCGACTGTGAGAAGTTG -3′
Control *(GFP)* 5′-GGGCGAGGAGCTGTTCACCG-3′

Constructs were packaged into lentivirus in HEK 293T cells using the Fugene HD and 2 µg of the vector, 2 µg psPAX2, and 1 µg pMD2-G. THP-1 cells supplemented with 10 µg/mL polybrene were spinfected with virus for 2 h at 1000g at 30 °C. After 2 days, cells were selected for stable expression of *S. pyogenes* Cas9 (Addgene #52962) using blasticidin (5 µg/mL for THP-1) and for stable expression of sgRNAs using puromycin (0.5 µg/mL for THP-1) or hygromycin (100 µg/mL for THP-1). After 10 d, single cells were isolated by serial dilution and expanded. Cells were then harvested for immunoblotting or used in experiments. CASP3, CASP7 and CASP9 single KOs were generated using this method.

## THP-1 knockout cell lines generated by electroporation of RNP particles

*In vitro* assembled ribonucleoproteins (RNPs) comprised of synthetic sgRNAs (IDT) and recombinant Cas9 protein (Alt-R *S.p.* HiFi Cas9 nuclease) (IDT) were electroporated into THP-1 monocytes as previously described using the Neon transfection system (Thermo Fisher) [20].

hCASP9 (sgrna sequence: 5'-CAACTTCTCACAGTCGATGT-3')
hCASP1 #1 (sgrna sequence: 5'-CGGCTTGACTTGTCCATTAT-3')
hCASP1 #2 (sgrna sequence: 5'-GACCTCTGACAGCACGTTCC-3')

Non-targeting (sgrna sequence: 5'-CGTTAATCGCGTATAATACG-3')

RNPs were assembled by mixing 0.3 µL of Cas9 protein (62 µM) with 0.5 µL of sgRNA (resuspended in nuclease-free water at a concentration of 100 µM) and incubated at RT for 15 min. $1.2 \times 10^6$ THP-1 cells were resuspended in 12 µL of T buffer, mixed with the assembled RNPs and electroporated using a 10 µL electroporation pipette tip with two 10 ms pulses at a voltage of 1400 V. Cells were then dispensed directly into a 6-well plate containing 3 ml of RPMI and cultured for 3–5 days before assessing bulk KO efficiency by immunoblot. WT and *CASP8/9* KO bulk THP-1 pools were further electroporated with CASP1 sgrna guides. Bulk KO efficiency was assessed again for target protein knockdown. These cell lines were further single cell cloned by limited serial dilution in RPMI to obtain clonal KO cell populations with complete knockdown of the target protein.

### Generation of HEK 293T CASP3/7 double KO cells

HEK 293T cells were transiently transfected with plasmids encoding for CAS9 and sgRNAs targeting CASP3 (sgRNA Sequence: AATGGACTCTGGAATATCCC) and CASP7 (sgRNA Sequence: GCATCTATCCCCCCTAAAGT). After 48, the cells were selected with puromycin (1 µg/mL) until all control cells were no longer viable. Single cell clones were then obtained by serial dilution and were confirmed by immunoblotting.

### Transient transfection

HEK 293T cells were seeded in 12-well culture plates at $0.25 \times 10^6$ cells/well in DMEM + 10% FBS. The following day, the indicated plasmids were mixed in Opti-MEM and transfected using FuGENE HD according to the manufacturer's protocol.

### LPS transfection of human macrophages

THP-1 cells were resuspended in RPMI medium + 10% FBS containing 50 ng/mL phorbol 12-myristate 12-acetate (PMA). Cells were plated in 96-well plates at a density of 8 x $10^4$ cells/well. After 48 h, the media was then replaced with Opti-MEM (0.1 mL/well). Where indicated, cells were treated with MCC950 (10 µM) for 30 min at 37 °C 5% $CO_2$ before LPS transfection. The LPS solution was prepared by adding LPS (25 mg/mL final concentration) and FuGENE (0.5% final concentration) to Opti-MEM. This solution was gently mixed by flicking and incubated for 30 min at room temperature before drop-wise addition to each well. The supernatants were collected separately, and cells were lysed by sonication 24 h post transfection. Harvested supernatants were precipitated by chloroform/methanol and analyzed by immunoblotting.

### HEK blue IL-1α/β reporter assay

HEK-Blue IL-1β (Invivogen) reporter cells were used according to manufacturer's protocols as previously described [21] to quantify the amount of active IL-1α/β released. HEK Blue cells were seeded in 96 well plates at 5 x $10^4$ cells/well in DMEM + 10% FBS to a total volume of 150 µL. 10 µL of supernatant from control or treated THP-1 cells were added, and the samples were incubated overnight at 37°C in a 5% $CO_2$ atmosphere incubator. A serial dilution of recombinant IL-1β (Invivogen) was included in all experiments to generate a standard curve which to calculate absolute levels of active IL-1β. 50 µL of supernatant was mixed with 150 µL of freshly prepared Quanti-Blue solution and incubated at room temperature for 10 minutes in the dark before reading absorbance at 620 nm using a Cytation 5 plate reader (Bio Tek).

### *Salmonella* infection of human macrophages

*Salmonella* Typhimurium SL1344 or SL1344 pDiGc [93] was streaked to isolation from frozen stock on LB + Streptomycin (100 µg/mL) agar plates. Carbenicillin (100 µg/mL) was included for all studies with SL1344 pDiGc. For SPI-I induced bacteria, single colonies were inoculated in LB + Streptomycin broth overnight (~18h). The overnight culture was back diluted 40-fold into LB + Streptomycin broth containing 300 mM NaCl. After three hours bacteria were washed 2x in sterile

PBS and cell density was determined by quantifying $OD_{600}$. MOI was prepared according to OD and bacterial cells were resuspended in PBS. 10 µL suspension of bacteria was added to macrophages in triplicate in 96-well plates. Plates were spun for 10 min at 400g to allow infection and then placed in incubator for 1 h. Media was then removed and replaced with fresh RPMI without FBS containing gentamicin (100 µg/mL). After 1 h, media was removed and wells were washed 3x with PBS and replaced with fresh RPMI without FBS containing gentamicin (10 µg/ml). GFP fluorescence was monitored using IncuCyte live cell imaging for 24 h.

For SL1344 (non-fluorescent) infection, after 1h 100 µg/mL gentamicin treatment, media was replaced with fresh RPMI without FBS containing gentamicin (10 µg/mL) and Sytox Green (0.5 µM). Cell death was quantified relative to 0.1% Triton treated macrophages.

For stationary phase bacterial infections, single colonies were inoculated in LB + Streptomycin for 2 days. MOI were prepared as described above and 10 µL suspension of bacteria were added to macrophages in triplicate in 96 well plates. Plates were spun for 10 min at 400g to allow infection and then placed in incubator for 2 h.

### *Salmonella* imaging and quantification

At 2 h or 24 h post *Salmonella* infection, media was replaced with fresh RPMI media without FBS containing Hoechst 33342 (Invitrogen) at 1:2000 dilution. Cells were incubated in the dark for 10 minutes prior to imaging using a 40x objective on a Leica DMI8 inverted widefield microscope equipped with Leica DFC9000 GT camera and LAS X software. Images were taken from triplicate wells. Images were processed in FIJI/ImageJ and bacteria in 150 infected cells were counted in all groups.

### LDH assays

Supernatants were harvested for LDH analyses at 24 h after bacterial infections and analyzed using the Cyquant LDH Cytotoxicity Assay (Thermo Scientific) according to the manufacturer's protocol. LDH activity was quantified relative to a lysis control where cells were lysed using 0.1% Triton-X.

### Western blotting

Protein samples were run on Nupage 4–12% Bis-Tris Midi Protein Gel (Thermo Scientific, WG1403BOX) at 175 volts. Proteins were transferred to 0.45 µm nitrocellulose membranes (BioRad,1704271) at 25 volts for 7 minutes using the Biorad Transblot Turbo System. Membranes were blocked using Intercept blocking buffer (LICORbio, 927–70010) for 1 hour and stained with primary antibodies at 1:1000 concentration in 50% Licor blocking buffer and 50% TBS buffer with 0.1% Tween for 1 hour at room temperature or overnight at 4° C. Membranes were washed 3x, followed by incubation with Donkey Anti-Mouse/Rabbit/Goat IgG Polyclonal Antibody (IRDye 800CW) for 1 hour at room temperature. Membranes were then washed 3x and imaged on the Odyssey M Imaging System (LICORbio). Images were analyzed with Empiria studio version 3.2 (LICORbio), and brightness, contrast, and tone parameters were adjusted uniformly in Adobe Photoshop.

### Statistical analysis

Data and statistical analysis were performed using GraphPad Prism 10 software. Statistical analysis was determined using One-way or Two-way ANOVA with multiple comparisons tests. Further details are listed under each figure legend.

### Supporting information

**S1 Fig. CASP1, CASP4, and CASP5 cleave CASP3 and CASP7 in cells. (A)** Confirmation of CASP3/7 knockout in HEK 293T cells by immunoblotting. **(B,C)** *CASP3*/*7* KO HEK 293T cells were transiently transfected with indicated constructs for 24 h then analyzed by immunoblotting. **(D-I)** HEK 293T cells stably expressing IL-18-V5 and ΔCARD

DmrB-CASP1, CASP4, or CASP5 were transiently transfected with indicated constructs coding for C-terminally Myc-tagged CASP3 (CASP3-Myc) **(D-F)** or C-terminally Flag-tagged CASP7 (CASP7-Flag) **(G-I)** for 24 h. Cells were then treated with 1 μM AP20187 for 5 h to activate the DmrB-caspases and cell lysates were analyzed by immunoblotting. Data are representative of three or more independent experiments.
(TIF)

**S2 Fig. CASP1, CASP8, and CASP9 are not required for CASP3/7 and GSDME processing during non-canonical inflammasome activation by intracellular LPS.** (**A-L**) Control-matched *CASP1* KO (**A-C**), *CASP8* KO (**D-F**), *CASP9* KO (**G-I**) and *CASP7* KO (**J-L**) THP-1 cells were transfected with 25 μg/mL LPS for 24 h. Supernatants and lysates were then analyzed by immunoblotting (**A, D, G, J**), cell death was measured by monitoring Sytox Green uptake (**B, E, H, K**), and CASP3/7 activity was determined using CellTreat CASP3/7 detection reagent (**C F, I, L**). Data are mean ± SEM of three independent replicates, and representative of at least three independent experiments. ****P < 0.0001, ***P < 0.001, **P < 0.01, and *P < 0.05 by two-way ANOVA test with Tukey's multiple comparison test comparing the control treated to KO treated samples at 24 h.
(TIF)

**S3 Fig. Stationary phase *Salmonella* induces modest CASP4/5 and CASP3-dependent activation of pyroptosis.** (**A-D**) WT and *CASP4/5* KO (**A,B**) or WT and *CASP3* KO (**C,D**) THP-1 macrophages were treated with the indicated MOI of WT Stm^Stat. for 24 h. Supernatants and lysates were analyzed by immunoblotting (**A,C**), and cell death was measured by monitoring Sytox Green uptake (**B,D**). Data are mean ± SEM of three independent experiments. ****P < 0.0001, ***P < 0.001, **P < 0.01, and *P < 0.05 by two-way ANOVA test with Tukey's multiple comparison test comparing the control treated to KO treated samples at 24 h.
(TIF)

**S4 Fig. CASP1, NLRC4, and NLRP3 are not required for CASP3/7 activation during *Salmonella* infection. (A-F)** WT and *CASP1* KO (**A-C**) or WT and *NLRC4* KO THP-1 macrophages (**D-F**) were treated with the indicated MOI of WT Stm^SPI-1 for 24 h. Supernatants and lysates were analyzed by immunoblotting (**A,D**), and cell death was measured by monitoring Sytox Green uptake (**B,E**), and CASP3/7 activity was determined using CellTreat CASP3/7 detection reagent (**C,F**). Data are mean ± SEM of three independent replicates, and representative of at least three independent experiments. ****P < 0.0001, ***P < 0.001, **P < 0.01, and *P < 0.05 by two-way ANOVA test with Tukey's multiple comparison test comparing the control treated to KO treated samples at 24 h.
(TIF)

**S5 Fig. PANX1 and NINJ1 contribute to *Salmonella*-induced cell death in GSDMD/E deficient macrophages. (A)** Examples of images of WT and *GSDMD/E* KO cells at MOI 20 used to quantify Sytox Green uptake are shown. White arrows show pyroptotic morphology and scale bars indicate 10 μm. **(B-D)** WT, *GSDMD* KO, and *GSDMD/E* KO THP-1 macrophages were infected with WT Stm^SPI-1 at MOI = 50 in the presence or absences of pretreatment with the PANX1 (20 μM spironolactone) or NINJ1 (50 mM glycine) inhibitors for 24 h then LDH release (**B**) and Sytox Green (**C,D**) were assessed. Data are mean ± SEM of three independent replicates, and representative of at least three independent experiments.****P < 0.0001, ***P < 0.001, **P < 0.01, and *P < 0.05 by one-way ANOVA test with Tukey's multiple comparisons test in B or two-way ANOVA test with Tukey's multiple comparison test comparing the control treated to KO treated samples at 24 h.
(TIF)

**S6 Fig. GSDMD and CASP3 are required, but GSDME is dispensable for limiting *Salmonella* replication in human macrophages.** (**A-D**) WT, *GSDMD* KO, *GSDME* KO and *GSDMD/E* KO THP-1 macrophages were treated with the indicated MOI of GFP-Stm^SPI-1. Representative images of the GFP-Stm^SPI-1 (MOI = 20) are shown in **A**. After 24 h, LDH assays

were performed (**B**). The kinetics of GFP-$Stm^{SPI-1}$ replication as measured by the fold change in green intensity is depicted in (**C**) and the quantification at 24 h is shown in **D**. Imaging was initiated 2 h post infection (t = 0) using an Incucyte and monitored for 24 h. (**E,F**) WT or *CASP3* KO THP-1 macrophages were treated with the indicated MOI of GFP-$Stm^{SPI-1}$. Representative images of the GFP-$Stm^{SPI-1}$ (MOI = 20) are shown in **E**. Quantification of the kinetics of the fold change in green intensity is depicted in **F**. Data are mean ± SEM of three independent replicates, and representative of at least three independent experiments. ****$P < 0.0001$, ***$P < 0.001$, **$P < 0.01$, and *$P < 0.05$ by one-way ANOVA test with Tukey's multiple comparison's test in B, D or two-way ANOVA test with Tukey's multiple comparison test comparing the control treated to KO treated samples at 24 h.
(TIF)

## Acknowledgments

We thank Dr. Derek Abbott for providing *GSDMD* KO, *GSDME* KO, *GSDMD/E* KO, and *CASP8* KO THP-1 cells. We thank Dr. Igor Brodsky for providing strain SL1344 pDiGc and Dr. Sunny Shin for providing THP-1 *NLRC4* KO cells for infection studies. We also thank Drs. Yi-Wei Chang and Sharon Patray for help with microscopy studies. The authors thank Emily Cento, Zhilin Chen, Max A. Eldabbas, and Emileigh Maddox of the Human Immunology Core and the Division of Transfusion Medicine and Therapeutic Pathology at the Perelman School of Medicine at the University of Pennsylvania for providing de-identified monocytes that were purified from healthy donor apheresis using StemCell RosetteSep kits. The HIC is supported in part by NIH P30 AI045008 and P30 CA016520. HIC RRID: SCR_022380.

## Author contributions

**Conceptualization:** Madhura Kulkarni, Christopher M. Bourne, Cornelius Y. Taabazuing.

**Data curation:** Madhura Kulkarni, Christopher M. Bourne, Cornelius Y. Taabazuing.

**Formal analysis:** Madhura Kulkarni, Christopher M. Bourne, Ashutosh B. Mahale, Patrick M. Exconde, Cecelia Murphy, Sofia Cervantes, Matilda Kardhashi, Mirai Kambayashi, William Yoo, Tristan J. Wrong, Robert C. Patio, Bohdana M. Discher, Cornelius Y. Taabazuing.

**Funding acquisition:** Christopher M. Bourne, Cornelius Y. Taabazuing.

**Investigation:** Madhura Kulkarni, Christopher M. Bourne, Ashutosh B. Mahale, Patrick M. Exconde, Cecelia Murphy, Haley T. Goodrow, Sofia Cervantes, Matilda Kardhashi, Mirai Kambayashi, William Yoo, Tristan J. Wrong, Robert C. Patio, Bohdana M. Discher.

**Methodology:** Madhura Kulkarni, Christopher M. Bourne, Cornelius Y. Taabazuing.

**Project administration:** Cornelius Y. Taabazuing.

**Resources:** Cornelius Y. Taabazuing.

**Supervision:** Cornelius Y. Taabazuing.

**Visualization:** Madhura Kulkarni, Christopher M. Bourne, Cornelius Y. Taabazuing.

**Writing – original draft:** Madhura Kulkarni, Christopher M. Bourne, Cornelius Y. Taabazuing.

**Writing – review & editing:** Madhura Kulkarni, Christopher M. Bourne, Cornelius Y. Taabazuing.

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
