## [Decision Letter · Decision Letter 0]

5 Jan 2026

PPATHOGENS-D-25-02867

Human non-canonical inflammasomes activate CASP3 to limit intracellular Salmonella replication in macrophages

PLOS Pathogens

Dear Dr. Taabazuing,

Thank you for submitting your manuscript to PLOS Pathogens. After careful consideration, we feel that it has merit but does not fully meet PLOS Pathogens's publication criteria as it currently stands. Therefore, we invite you to submit a revised version of the manuscript that addresses the points raised during the review process.

We look forward to receiving your revised manuscript.

Kind regards,

Dana J. Philpott

Academic Editor

PLOS Pathogens

Eva Heinz

Section Editor

PLOS Pathogens Sumita Bhaduri-McIntosh

Editor-in-Chief

PLOS Pathogens

orcid.org/0000-0003-2946-9497

Michael Malim

Editor-in-Chief

PLOS Pathogens

orcid.org/0000-0002-7699-2064

**Additional Editor Comments:**

Thank you for submitting your work to PLOS Pathogens. The Reviewers were quite positive about the work but have some concerns that will need to be addressed. In particular, there were comments related to the timing of cleavage of substrates in the Western blots and recommendations to validate some of the work with recombinant proteins to prove specificity. Please provide a detailed point-by-point response to each comment.

**Journal Requirements:**

At this stage, the following Authors/Authors require contributions: Madhura Kulkarni, Christopher Bourne, Ashutosh Mahale, Patrick Exconde, Cecelia Murphy, Sofia Cervantes, Matilda Kardhashi, Mirai Kambayashi, William Yoo, Tristan Wrong, Robert Patio, Bohdana Discher, and Cornelius Taabazuing. Please ensure that the full contributions of each author are acknowledged in the "Add/Edit/Remove Authors" section of our submission form.

- ® on page: 16 and 20.

- TM on page: 16 and 21..

5) We have noticed that you have uploaded Supporting Information files, but you have not included a list of legends. Please add a full list of legends for your Supporting Information files after the references list.

6) In the online submission form, you indicated that "Further information and requests for resources and reagents should be directed to and will be fulfilled by the lead contact, Cornelius Y. Taabazuing (Cornelius.taabazuing@pennmedicine.upenn.edu)". All PLOS journals now require all data underlying the findings described in their manuscript to be freely available to other researchers, either

1. In a public repository

2. Within the manuscript itself

3. Uploaded as supplementary information.

7) Please amend your detailed Financial Disclosure statement. This is published with the article. It must therefore be completed in full sentences and contain the exact wording you wish to be published.

**Reviewers' Comments:**

Reviewer's Responses to Questions

**Part I - Summary**

Reviewer #1: This paper is a detailed analysis of the role of non-canonical inflammasomes in activating CASP3. The important result of the paper is that in human cells, CASP4/5 can cleave CASP3. CASP3 seems to play a small role in restriction of Salmonella replication but how it does so remains unclear. Overall I feel the paper will be of some interest to specialists interested in the details of pathogen-induced cell death.

Reviewer #2: In this study, the authors explore the links between inflammatory caspases and their substrates in the control of Salmonella replication. The data presented combines biochemistry and genetics in an impressive manner, leading to simple interpretation of the data presented. As Salmonella induced cell death, and its impact on intracellular replication, have been heated topics of investigation for many years, this study will likely attract significant attention from the community. With this expectation in mind, the following suggestions are offered to improve the quality of the data supporting the conclusions offered. Point 2 is the most important to address.

1. Line 80 describes the authors report that caspase-4/5 cleaves pro-IL-1 and pro-IL-18. The author should cite the prior studies by Shao and Kagan on this topic.

2. Although the caspase substrate cleavage assays presented throughout this study are impressively clean (with appropriate controls), the length of time used in these assays atypical. The 1-3 hours of cleavage time is not appropriate for the authors to establish the efficiency of cleavage of each substrate of interest. The authors are encouraged to perform assays using recombinant proteins, as has been established by the labs indicated in point 1 above, in order to determine the relative efficiency by which caspase-7 is cleaved. This suggestion, while important for the authors are consider, should not be taken as a requirement to replace all assays in this study with those using recombinant proteins. However, some key experiments would be important to include in a revised manuscript.

3. Figure 3A is confusing, as the authors show that caspase-4/5 double knockout cells retain the ability to cleave GSDMD after LPS treatment. As current literature suggests that caspase-4/5 would be the only pathway to GSDMD cleavage, the authors are encouraged to explore this observation further.

4. The acronym Salmonella enterica serovar Typhimurium (Stm) is described multiple times in this manuscript. Please adjust for one description.

Reviewer #3: The study by Kulkarni et al. used human macrophages to characterize the role played by CASP4/5 in the cleavage of CASP3, which in turn leads to the cleavage of GSDME, and this pathway was functional in response to both intracellular LPS as well as infection with Salmonella.

Overall, this is an interesting and carefully performed study that clarifies the contributions of GSDMD-dependent pyroptosis versus CASP3-dependent functions (either apoptosis or GSDME-dependent pore formation) in human macrophages.

**Part II – Major Issues: Key Experiments Required for Acceptance**

Please use this section to detail the key new experiments or modifications of existing experiments that should be absolutely required to validate study conclusions.required to validate study conclusions.

Reviewer #1: In terms of concrete suggestions to improve the manuscript, I think there are two major issues that should be addressed:

1. The claim that CASP3/7 are “direct” substrates of CASP4/5 needs experiments in which both the enzyme (CASP4 or 5) and the substrate (CASP3 or 7) are purified recombinant proteins. Unless I missed something, the experiments performed used cell lysates expressing CASP3/7 instead of purified CASP3/7. I believe that the presence of other proteins in the lysate creates the possibility that cleavage of CASP3/7 is indirect. In addition, some enzymology (kcat/Km) to show that the cleavage is efficient (comparable to other bona fide substrates) would bolster the case that these are indeed truly direct substrates and not simply being cleaved during non-physiological experimental conditions. I also believe that the cleavage site should be identified, either by MS or by mutating the site and showing cleavage no longer occurs.

2. The claim that CASP1/8/9 are dispensable for CASP3/7 activation needs to be confirmed in CASP1/8/9 triple KO cells. Although individual KO are tested, the possibility remains that there is compensation/redundancy.

Reviewer #2: (No Response)

Reviewer #3: In Fig. 3A, there is still a relatively substantial cleavage of GSDMD observed, as well as CASP1 p10 formation, in CASP4/5 DKO cells in response to intracellular LPS. Given that CASP4/5 are the sensors of intracellular LPS, how can this be explained? Is this processing driven by TLR4? Using TLR4 KO cells would clarify this point.

The authors used Salmonella grown to stationary phase to “to minimize the contribution of the canonical caspase-1 signaling pathway” (line 258). This is not very clear and the authors should explain what the growth conditions is expected to do. Is it only to limit expression of the TTSS and thereby NLRC4 detection? Or also to limit NLRP3? Or both? But isn’t the Salmonella strain used flagellated? This would likely also drive NLRC4. Using flagellated versus non-flagellated strains could solve this point.

In Fig 5A-B, the results obtained with GSDME KO macrophages are puzzling. I wonder if different results would have been obtained if the initial time against which everything is measured was earlier than 2h… Indeed, 2h is already a very long time when considering pyroptotic events, which occur in minutes.

**Part III – Minor Issues: Editorial and Data Presentation Modifications**

Reviewer #1: Minor points:

line 18 (abstract) -- reword so as not to imply that cleavage of IL-1/18 induces pyroptosis

line 24-26 (abstract) -- I find this sentence to be confusing. It is not obvious to me how it follows from the fact that CASP1 is responsible for GSDMD cleavage that GSDMD not GSDME is the principal driver of pyroptosis

Line 140 - Fig. 1F (not Fig. 1C)

Figure 3B/E/H/K -- why does cell death of control cells reach 100% in some experiments, but only 25-30% in others?

Figure 4 -- why do GSDME KO cells die more in Salmonella infection than controls? This deserves some explanation, especially given that in Fig 3, the GSDME KO cells are defective for cell death in response to LPS.

Reviewer #2: (No Response)

Reviewer #3: Lines 105-6: I don’t understand why the lack of human CASP3 cleavage by CASP11 “aligns” with data on the cleavage of mIL18. It just looks like two totally unrelated facts.

Line 161: Is the effect of MCC950 on CASP3/GSDME driven by the lack of activation of CASP1 or the lack of activation of GSDMD?

Lines 241-48: what about the release of active IL18?

Line 460: “if GSDME pores are sublytic, as has been suggested, it could facilitate pathogen entry” I think the authors should clarify what they mean here. One may think that this would suggest that the bacteria could enter through the GSDME pores, which is certainly not realistic.

PLOS authors have the option to publish the peer review history of their article (what does this mean?). If published, this will include your full peer review and any attached files.). If published, this will include your full peer review and any attached files.

.

Reviewer #1: No

Reviewer #2: No

Reviewer #3: No

**Figure resubmission:**
---

## [Editor Report · Decision Letter 1]

16 Apr 2026

Dear Dr. Taabazuing,

We are pleased to inform you that your manuscript 'Human non-canonical inflammasomes activate CASP3 to limit intracellular Salmonella replication in macrophages' has been provisionally accepted for publication in PLOS Pathogens.

Best regards,

Dana J. Philpott

Academic Editor

PLOS Pathogens

Eva Heinz

Section Editor

PLOS Pathogens

Sumita Bhaduri-McIntosh

Editor-in-Chief

PLOS Pathogens

orcid.org/0000-0003-2946-9497

Michael Malim

Editor-in-Chief

PLOS Pathogens

orcid.org/0000-0002-7699-2064

Thank you for addressing the reviewers' comments - very well done and comprehensive.
---

## [Editor Report · Acceptance letter]

Dear Dr. Taabazuing,

We are delighted to inform you that your manuscript, "Human non-canonical inflammasomes activate CASP3 to limit intracellular Salmonella replication in macrophages," has been formally accepted for publication in PLOS Pathogens.

Best regards,

Sumita Bhaduri-McIntosh

Editor-in-Chief

PLOS Pathogens

orcid.org/0000-0003-2946-9497

Michael Malim

Editor-in-Chief

PLOS Pathogens

orcid.org/0000-0002-7699-2064